# Methylated *cis*-regulatory elements mediate KLF4-dependent gene transactivation and cell migration

Jun Wan[1†], Yijing Su[2,3], Qifeng Song[4,5], Brian Tung[2,6], Olutobi Oyinlade[4], Sheng Liu[1], Mingyao Ying[2,6], Guo-li Ming[2], Hongjun Song[2], Jiang Qian[1,2,3,7]*, Heng Zhu[2,4,5,6]*, Shuli Xia[2]*

[1]The Wilmer Eye Institute, Johns Hopkins University School of Medicine, Baltimore, United States; [2]Department of Neurology, Johns Hopkins University School of Medicine, Baltimore, United States; [3]Institute for Cell Engineering, Johns Hopkins University School of Medicine, Baltimore, United States; [4]Department of Pharmacology and Molecular Sciences, Johns Hopkins University School of Medicine, Baltimore, United States; [5]Center for High-Throughput Biology, Johns Hopkins University School of Medicine, Baltimore, United States; [6]Hugo W Moser Research Institute at Kennedy Krieger, Baltimore, United States; [7]The Solomon Snyder Department of Neuroscience, Johns Hopkins University School of Medicine, Baltimore, United States

*For correspondence: jiang.qian@ jhmi.edu (JQ); hzhu4@jhmi.edu (HZ); xia@kennedykrieger.org (SX)

Present address: [†]Department of Medical and Molecular Genetics, Indiana University School of Medicine, Indianapolis, United States

Competing interests: The authors declare that no competing interests exist.

**Abstract** Altered DNA methylation status is associated with human diseases and cancer; however, the underlying molecular mechanisms remain elusive. We previously identified many human transcription factors, including Krüppel-like factor 4 (KLF4), as sequence-specific DNA methylation readers that preferentially recognize methylated CpG (mCpG), here we report the biological function of mCpG-dependent gene regulation by KLF4 in glioblastoma cells. We show that KLF4 promotes cell adhesion, migration, and morphological changes, all of which are abolished by R458A mutation. Surprisingly, 116 genes are directly activated via mCpG-dependent KLF4 binding activity. In-depth mechanistic studies reveal that recruitment of KLF4 to the methylated *cis*-regulatory elements of these genes result in chromatin remodeling and transcription activation. Our study demonstrates a new paradigm of DNA methylation-mediated gene activation and chromatin remodeling, and provides a general framework to dissect the biological functions of DNA methylation readers and effectors.

## Introduction

DNA methylation at the five position of the cytosine base (5mC) is the primary epigenetic modification on the mammalian genomic DNA (*Jaenisch and Bird, 2003*), and dysregulation of DNA methylation is a hallmark of various diseases and cancer (*Sharma et al., 2010*). CpG methylation in *cis*-regulatory elements is generally believed to repress gene expression by disrupting transcription factor (TF)-DNA interactions directly or indirectly via the recruitment of proteins containing methyl-CpG-binding domain (MBD), which are largely sequence independent (*Boyes and Bird, 1991*). This dogma has been challenged by several recent studies in which many TFs were identified as a new class of sequence-specific methylated DNA readers (*Filion et al., 2006*; *Mann et al., 2013*; *Rishi et al., 2010*; *Sasai et al., 2010*; *Serra et al., 2014*; *Spruijt et al., 2013*; *Zhu et al., 2016*). For example, a large-scale survey against the human TF repertoire revealed that 47 TFs and co-factors,

including the Krüppel-like factor 4 (KLF4), recognize mCpG-containing DNA motifs in a sequence-specific manner (*Hu et al., 2013*). In addition to its canonical DNA motif, KLF4 also recognizes a different motif that requires CpG methylation (i.e., 5'-CmCGC). Further mutagenesis studies demonstrated that Arg458-to-Alanine (R458A) mutation in KLF4 abolished its interaction with the methylated DNA motif, but showed no detectable impact on binding to its canonical, unmethylated motifs (*Hu et al., 2013*). Intriguingly, cell-based luciferase assays showed that KLF4 wild type (WT) could recognize methylated promoter and activate downstream transcription; while R458A mutant could not (*Hu et al., 2013*). The crystal structure of KLF4 binding to mCpGs was published by an independent study (*Liu et al., 2014*). However, the physiological function of the mCpG-dependent KLF4 binding activity in mammalian cells remains unknown.

KLF4 plays multiple roles in normal physiology and disease. It is one of the Yamanaka factors that induce pluripotency in somatic cells (*Nandan and Yang, 2009*). KLF4 also functions as a cancer driver gene (*Vogelstein et al., 2013*), and is involved in cancer stem cell maintenance (*Leng et al., 2013*; *Yu et al., 2011*; *Zhu et al., 2014*). For example, Our previous studies indicate that treatment of cancer cells with hepatocyte growth factor induced cancer stem cell phenotypes by increasing the expression of reprograming factors including KLF4 (*Li et al., 2011*). KLF4 has also been shown to be upregulated in high-grade brain tumors (*Elsir et al., 2014*; *Holmberg et al., 2011*), such as glioblastoma (GBM), the most aggressive and lethal adult brain tumor (*Carlsson et al., 2014*; *Quick et al., 2010*). In addition to driving tumor malignancy, KLF4 can act as a tumor suppressor in distinct cellular contexts (*Evans and Liu, 2008*; *Rowland et al., 2005*; *Tetreault et al., 2013*).

In this study, we dissected the biological function of KLF4 binding to methylated DNA in malignant brain tumor cells by taking advantage of the R458A mutant lacking the ability to bind to methylated DNA. Our study showed that KLF4-mCpG interaction promotes brain tumor cell migration via the transactivation of genes involved in cell motility pathways, including the small GTPase *RHOC*. We further demonstrate that recruitment of KLF4 to methylated *cis*-regulatory elements results in chromatin remodeling and activation of gene transcription in a genome-wide scale.

## Results

### KLF4-mCpG interaction promotes GBM cell adhesion and migration

We chose two human GBM cell lines, namely U87 and U373, with physiologically relevant genetic backgrounds to dissect the function of mCpG-dependent KLF4 binding activity, and because these two cell lines have low endogenous KLF4 expression. To facilitate in vivo studies of mCpG-dependent binding activity of KLF4, we engineered two stable cell lines that, upon doxycycline (Dox) treatment, each expressed KLF4 WT or KLF4 R458A. Cells without doxycycline treatment served as a negative control. Western blot analysis and immunocytochemistry staining confirmed that the endogenous KLF4 level in non-transfected U87 cells was barely detectable; after 48 hr Dox induction, both KLF4 WT and R458A proteins showed a dose-dependent increase without an impact on cell proliferation (*Figure 1A*, *Figure 1—figure supplement 1A,B*). For the rest of the study, we chose the Dox dose 1 µg/ml, which induced KLF4 expression to a level (~20–27 fold) similar to that during cancer cell reprogramming when challenged by growth factors (*Li et al., 2011*).

We observed that induction of KLF4 WT significantly increased cell adhesion (*Figure 1B,C*) and promoted migration in both transwell assays and wound healing assays (*Figure 1D–F*). Similar results were obtained at 24- and 48 hr post induction (*Figure 1—figure supplement 1C*). In contrast, induction of R458A had no detectable impact on either cell adhesion or migration (*Figure 1B–F*). Consistent with the observed phenotypes, KLF4 WT-expressing cells showed elongated, spindle-like morphology (arrows, *Figure 2A*), whereas control and R458A-expressing cells remained round and small (arrowheads, *Figure 2A*). Quantitative analysis further confirmed this observation (*Figure 2B*). Immunofluorescence staining for F-actin and vinculin further confirmed the formation of stress actin fibers and focal adhesion, respectively, in KLF4 WT-expressing GBM cells (arrowheads), but not in R458A-expressing cells (*Figure 2C*). Similar results were observed in other GBM cell lines, such as U373 GBM cells (*Figure 2—figure supplement 1A–D*). Since R458A loses binding activity to mCpG-containing motifs but retains binding to unmethylated canonical motifs (*Hu et al., 2013*), these phenotypic differences can be attributed to the mCpG-dependent KLF4 activity.

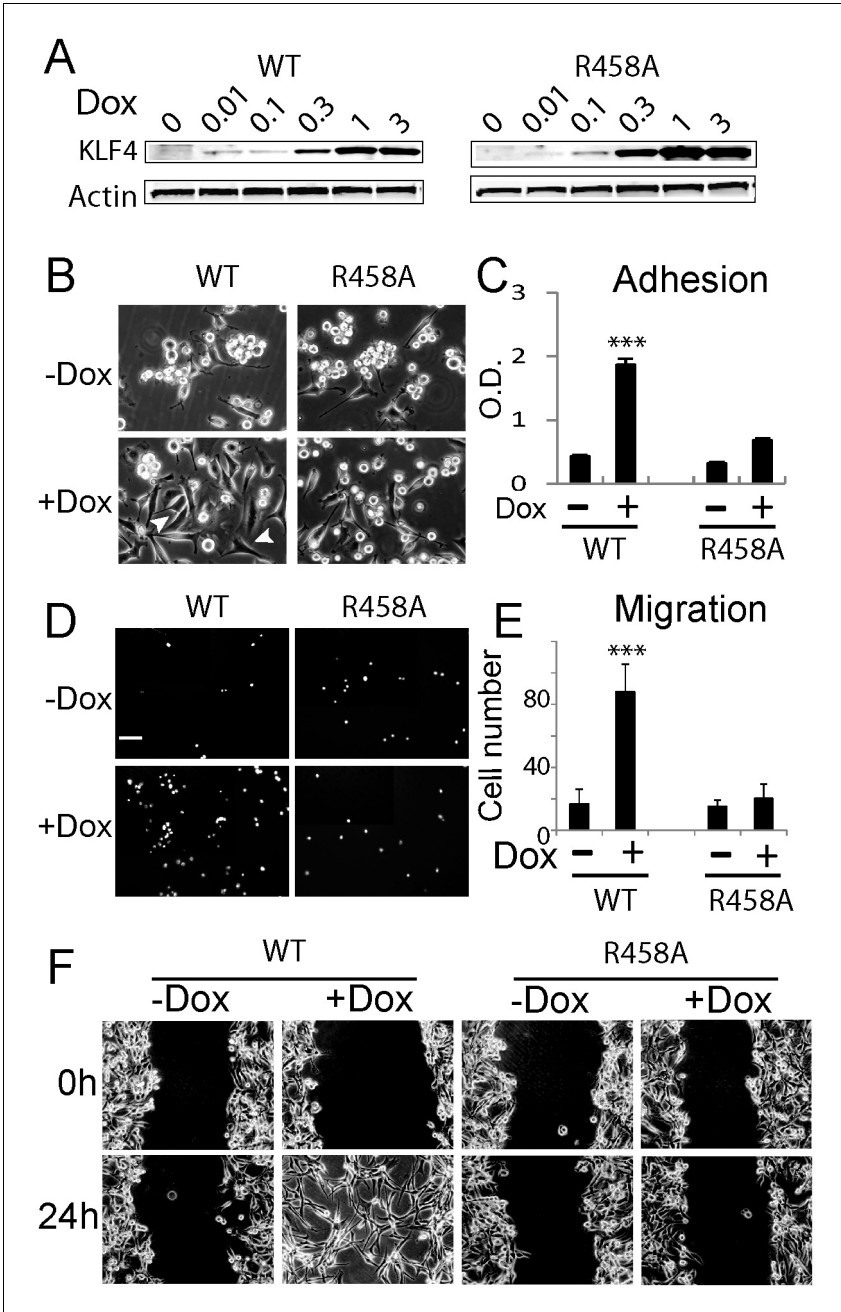

**Figure 1.** Methyl CpG-dependent KLF4 binding activity promoted GBM cell adhesion and migration. (**A**) Induced expression of KLF4 WT and R458A in human U87 GBM cells upon doxycycline (Dox) treatment. Cells were transfected with lentivirus harboring tet-on KLF4 WT or R458A constructs and selected with antibiotics. Stable cell lines were treated with Dox for 48 hr before immunoblotting. Immunoblotting showing induced expression of KLF4 WT and R458A in human U87 GBM cells upon doxycycline (Dox) treatment. (**B and C**) KLF4 WT but not R458A promoted GBM cell adhesion. Cells were pre-treated with Dox for 48 hr and plated for 2 hr before washing. (**D and E**) KLF4 WT promoted GBM cell migration in broyden chamber transwells. Cells were pretreated with Dox for 48 hr, plated on transwells containing 0.1% FCS and migrating towards 10% FCS. After 3 hr, migrating cells were stained with DAPI and five field / transwell were counted. (**F**) KLF4 WT but not R458A promoted GBM cell migration in wound healing assays. Cells were treated with Dox for 5 days till confluence. A scratch was made and cells were maintained in 0.1% FCS medium overnight, cell proliferation was inhibited by mitomycin C. Microphotographs were taken 0 hr and 24 hr after scratching. Bar = 25 µm. ***p<0.001.

The following figure supplement is available for figure 1:

*Figure 1 continued on next page*

*Figure 1 continued*

**Figure supplement 1.** Expression of KLF4 WT and R458A mutant did not affect cell growth.

To confirm that the observed phenotypes were DNA methylation dependent, we pre-treated the cells with a DNA methytransferase inhibitor, 5-aza-2′-deoxycytidine (5-Aza, 1µmol/L), which eliminates DNA methylation on a genome-wide scale (*Chen et al., 1998*). As illustrated in *Figures 2D,E*, 5-Aza treatment partially suppressed the enhanced cell migration and wound healing phenotypes driven by KLF4 WT, suggesting that the increased cell migration was indeed mediated via a DNA methylation-dependent mechanism.

## Identification of transcriptional network regulated by KLF4-mCpG interactions

The distinct phenotypes induced specifically by KLF4 WT suggested that mCpG-dependent KLF4 binding events regulate the expression of genes involved in these biological processes. To elucidate the molecular mechanisms by which KLF4 regulates gene transcription via its mCpG-dependent activity, we carried out a series of genome-wide analyses to identify the direct gene targets that are regulated by KLF4 WT but not R458A.

First, we performed RNA-seq analysis before (0 hr) and after (48 hr) induction of KLF4 WT and R458A, respectively. Approximately 86% of the sequencing reads were uniquely mapped to the human genome (*Figure 3—source data 1*); comparison of the expression profiles between the replicates showed high reproducibility (*Figure 3—figure supplement 1A*). Differentially expressed genes (DEGs) after KLF4 induction were identified by comparing gene expression at 48 hr versus 0 hr of KLF4 expression. We observed that a total of 613 genes were significantly up- or down-regulated post KLF4 WT induction (p<0.001), indicating that KLF4 globally affected gene transcription (*Figure 3A*). Among them, a large fraction (500/613 = 82%) was only affected by KLF4 WT but not R458A, suggesting that these expression changes were regulated, directly or indirectly, via mCpG-dependent KLF4 binding activity (*Figure 3B*). Surprisingly, 308 of the 500 genes showed significantly elevated expression after KLF4 WT induction (*Figure 3B*). For example, the expression of *RHOC*, *RAC1*, *LMO7*, and *MIDN* was up-regulated by induction of KLF4 WT but not R458A (*Figure 3C*). This result suggested that mCpG-dependent KLF4-binding could activate cellular gene transcription and therefore, we decided to focus on these activated genes in the rest of our study.

To determine which genes were directly activated by mCpG-dependent KLF4 binding events, we next performed genome-wide chromatin immunoprecipitation-sequencing (ChIP-seq) in KLF4 WT and R458A-expressing cells (i.e., 48 hr post induction). At least 70% of the ChIP-seq reads were mapped to the human genome (*Figure 3—source data 2*). A total of 3890 and 1222 significant ChIP-seq peaks were identified in KLF4 WT and R458A expressing cells, respectively (*Figure 3D*). A comparison between the KLF4 WT and R458A ChIP-seq peaks identified that 2733 (70%) were specific to KLF4 WT, indicating that these peaks were recognized via mCpG-dependent KLF4 binding activity (referred as WT-specific peaks) (*Figure 3D*). In contrast, ~95% of the KLF4 R458A ChIP-seq peaks were also recognized by KLF4 WT (referred as shared peaks), indicating that a single R458A mutation abolished >2/3 of the KLF4 WT binding loci in the chromatins (*Figure 3—source data 3*). Sequence reads distribution of KLF4 WT and R458A ChIP-seq peaks at the promoter region of *RHOC*, as well as mapped *RHOC* RNA-seq, are shown in *Figure 3E* as an example. More examples can be found in *Figure 3—figure supplement 1B,C*.

To fully examine the DNA methylation status of the WT and R458A ChIP-seq peaks, we performed whole genome bisulfite sequencing to decode the methylome of U87 cells and combined the DNA methylome data separately with the KLF4 WT and R458A ChIP-seq datasets. We found that 66% of the KLF4 WT-specific ChIP-seq peaks showed a high methylation level (e.g., $\beta$ >60%) at CpG sites, while only 36% of the ChIP-seq peaks shared by KLF4 WT and R458A reached a similar CpG methylation level (p=3.7e-223). Different cutoffs for defining high methylation levels did not alter this observation (*Figure 3F*). Therefore, the KLF4 WT-specific ChIP peaks are enriched for highly methylated CpGs.

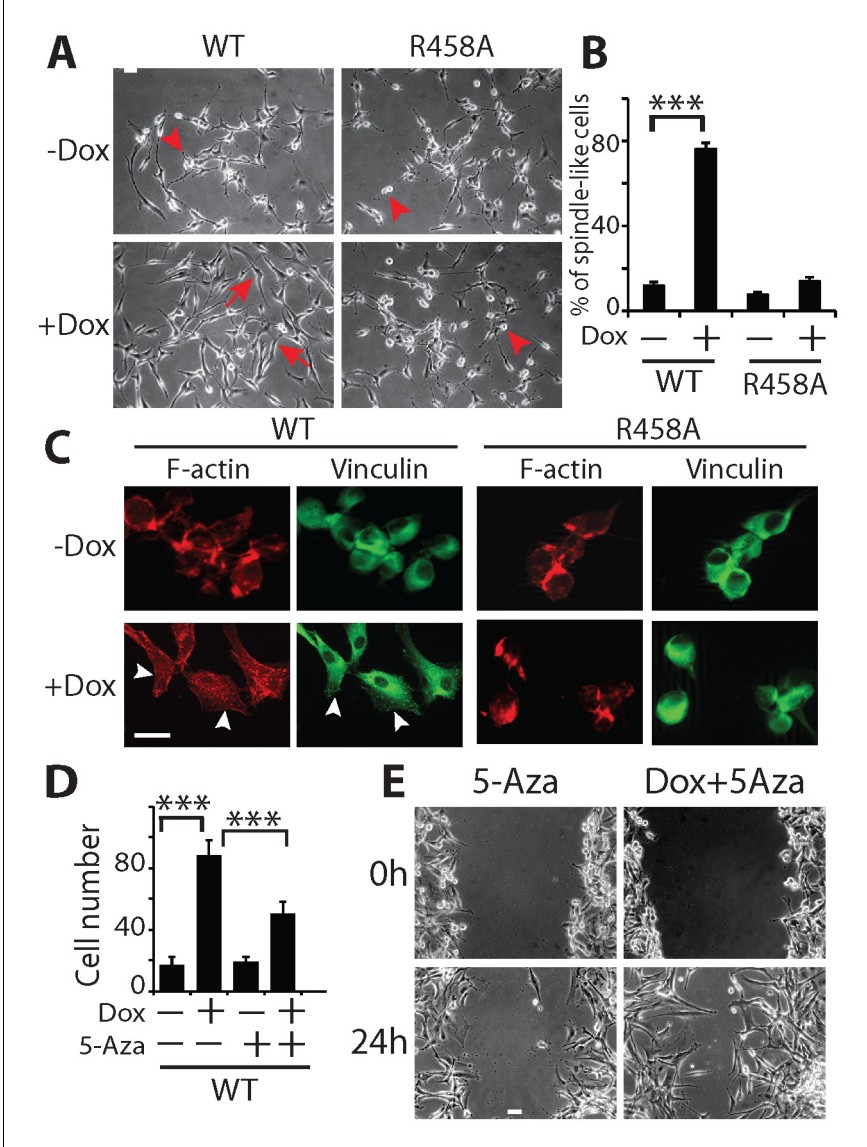

**Figure 2.** KLF4 WT-induced cell migration is methylation dependent. (**A**) Expression of KLF4 WT but not R458A induced cell morphology changes in U87 cells (+Dox, 48 hr). The control and R458A expressing cells showed round and short cell body (arrowheads), whereas KLF4 WT induced elongated, spindle-like cell shape (arrows). Bar = 25 μm. (**B**) Quantification of cell morphology changes after Dox treatment for 48 hr. More than 80% of the KLF4 WT expressing cells showed elongated, spindle-like cell morphology, whereas most KLF4 R458A expressing cells remained as round and short cells. (**C**) F-actin and vinculin staining in control, KLF4 WT and R458A-expressing cells. KLF4 WT induced actin stress fiber formation and focal adhesion formation (arrowheads). Bar = 10 μm. (**D**) Pre-treatment of the cells with DNA methyltransferase inhibitor 5-Aza (1μmol/L, 10 days) blocked KLF4 WT-induced cell migration in transwell assays. (**E**) Scratch assays indicating that 5-Aza blocked wound healing induced by KLF4 WT. Bar = 25 μm. ***p<0.001.

The following figure supplement is available for figure 2:

**Figure supplement 1.** Methyl CpG-dependent KLF4 binding activity promoted adhesion and migration of human U373 GBM cells.

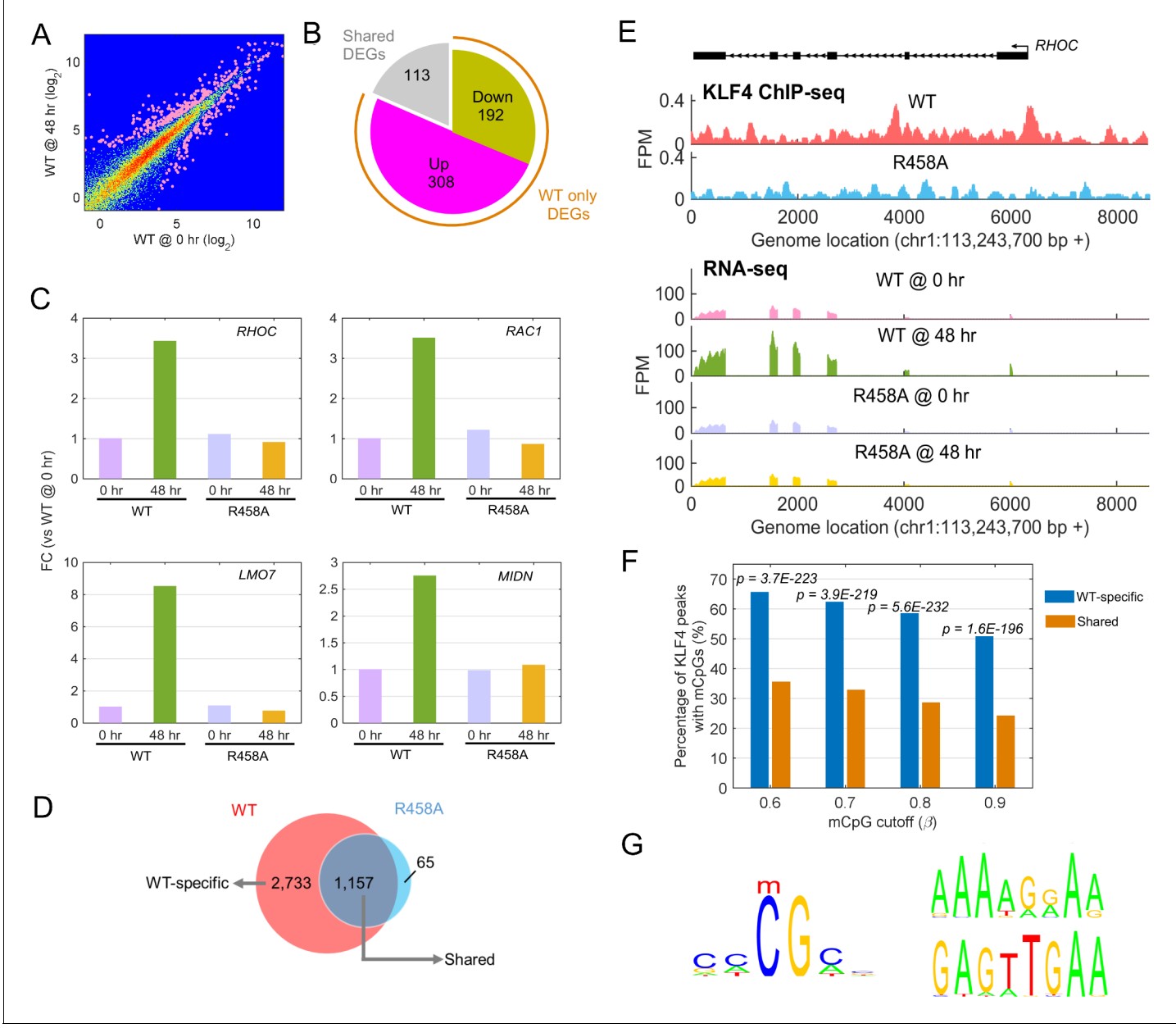

**Figure 3.** Identify transcriptional network regulated by KLF4-mCpG interactions. (**A**) RNA-seq data before (0 hr) and after (48 hr) KLF4 WT induction. The pink dots were determined as differential expressed genes (DEGs) (p<0.001). (**B**) Overlap between DEGs in KLF4 WT and R458A cells, showing that a total of 613 genes were significantly regulated by KLF4 WT, 115 of which were also significantly regulated by KLF4 R458A. Among the rest 500 genes significantly regulated by KLF4 WT but not R458A (WT only DEGs), 308 of them were up-regulated by KLF4 WT only. (**C**) Four examples of KLF4 WT only DEGs. (**D**) Overlap between KLF4 WT and R458A KLF4 ChIP-seq peaks (48 hr +Dox), indicating that ~2733 peaks can be only bound by KLF4 WT; ~1157 peaks bound by both KLF4 WT and R458A, whereas R458A alone only bound a few new sites. (**E**) ChIP-Seq for KLF4 WT and R458A on and surrounding *RHOC* promoter as an example. RNA-seq at the same region was also shown, pre and post KLF4 WT and R458A induction, respectively. (**F**) Percentage of ChIP-seq peaks with mCpGs evaluated by whole genome bisulfite sequencing analysis. A significant enrichment was observed for methylated CpG in KLF4 WT-specific peaks (Blue bar) as compared to KLF4 R458A shared peaks (orange bar). (**G**) Motifs identified for KLF4-mCpG binding in KLF4 WT-specific peaks (Left) and for KLF4 R458A shared peaks (Right), respectively.

The following source data and figure supplement are available for figure 3:

**Source data 1.** Mapped reads for all the RNA-sequencing experiments.

**Source data 2.** Mapped reads for the ChIP-sequencing experiments.

*Figure 3 continued*

**Source data 3.** Chromosol location of KLF4 WT-specific, shared, and mutant-specific peaks.
**Source data 4.** Methylated 6-mer cis motifs identified in KLF4 WT-specific peaks.
**Figure supplement 1.** Analysis of RNA-seq and ChIP-seq data.

Next, we carried out motif analysis to identify enriched and highly methylated 6-mer DNA motifs in WT-specific ChIP-seq peaks, as well as in shared ChIP-seq peaks. At a cutoff of $\beta$ >60% CpG methylation we found 10 methylated 6-mer motifs (*Figure 3—source data 4*) that were significantly over-represented in the WT-specific peaks (p=6.6e-37). Many of them share sequence similarity to the motif 5'-CCCGCC (*Figure 3G*; left panel), of which the methylated form was reported to be recognized by KLF4 in our previous study (*Hu et al., 2013*). In contrast, the peaks shared by WT and R458A were found enriched for different motifs (e.g., 5'-AAAAGGAA and 5'- GAGTTGAA) (*Figure 3G*; right panel). Taken together, these results confirmed that the KLF4 WT-specific ChIP-seq peaks were enriched for highly methylated KLF4 binding motifs.

## Identification of direct targets of mCpG-dependent KLF4 interactions in GBM cells

To identify genes that were directly activated via mCpG-mediated KLF4 binding activity, we first searched the 2,733 KLF4 WT-specific ChIP-seq peaks against the genomic locations on those proximal regulatory regions, which were classified into three categories: upstream (~10 kb upstream to transcription start sites), 5'-UTRs, and exons. The proximal regulatory regions of 65 KLF4 WT up-regulated genes were found to be occupied by KLF4 WT-specific ChIP-seq peaks, indicating that they were direct targets of KLF4-mCpG interactions (*Supplementary file 1*).

We also noticed that most of the 2733 KLF4 WT-specific ChIP-seq peaks were located outside the proximal regulatory regions, suggesting that KLF4 might also activate gene expression via binding to distal enhancers. Therefore, we performed anti-H3K27ac ChIP-seq analysis and combined the obtained H3K27ac peaks with KLF4 WT-specific binding sites to identify the potential enhancer regions bound by KLF4 WT. 1773 out of 2733 KLF4 WT-specific ChIP-seq peaks overlapped with the 27,997 H3K27ac ChIP-seq peaks (64.5%) (*Figure 4A*). Using an enhancer target prediction algorithm that connects enhancers to specific genes (enhanceratlas.org) (*Gao et al., 2016*; *He et al., 2014*), we identified 51 additional genes that were up-regulated via mCpG-dependent KLF4 binding events (*Supplementary file 1*). Therefore, the up-regulation of 116 genes was found directly associated with mCpG-dependent KLF4 binding to their *cis*-regulatory elements in the proximal regulatory regions (56%) and distal enhancers (44%) (*Figure 4B* and *Supplementary file 1*).

We next examined whether this association was statistically significant. We first focused on 12,824 genes that were expressed (FPKM >0.5) in U87 cells, among which 308 (2.4%) genes were found up-regulated only in KLF4 WT-induced cells. Meanwhile, we observed that KLF4 mCpG-dependent binding peaks were associated with 2518 expressed genes, of which 116 (4.6%) were up-regulated genes that is significantly higher than the ratio of expressed genes not associated with KLF4 WT binding sites (fold enrichment = 1.9; p=5.0e-13).

To directly assess the methylation status in the *cis*-regulatory elements of the 116 direct target genes, we integrated the methylome dataset and found that the majority (72%) of the 116 genes were indeed associated with highly methylated ($\beta$ > 0.6) *cis*-regulatory elements (*Figure 4C*). To confirm this observation, we randomly selected 15 loci and performed methylation analysis by Sanger sequencing. Ten of them were confirmed to contain highly (75–100%) methylated CpG sites in the associated *cis*-regulatory elements. A few examples are shown in *Figure 4D* and *Figure 4—figure supplement 1*.

Gene Ontology analysis showed that these 116 direct targets of KLF4 WT were significantly enriched in biological processes relevant to the observed phenotypes, such as cell communication, regulation or establishment of localization, cell adhesion or morphogenesis, positive regulation of MAP kinase activity, cell-cell junction, and cytoskeleton organization (*Figure 4E*). These results

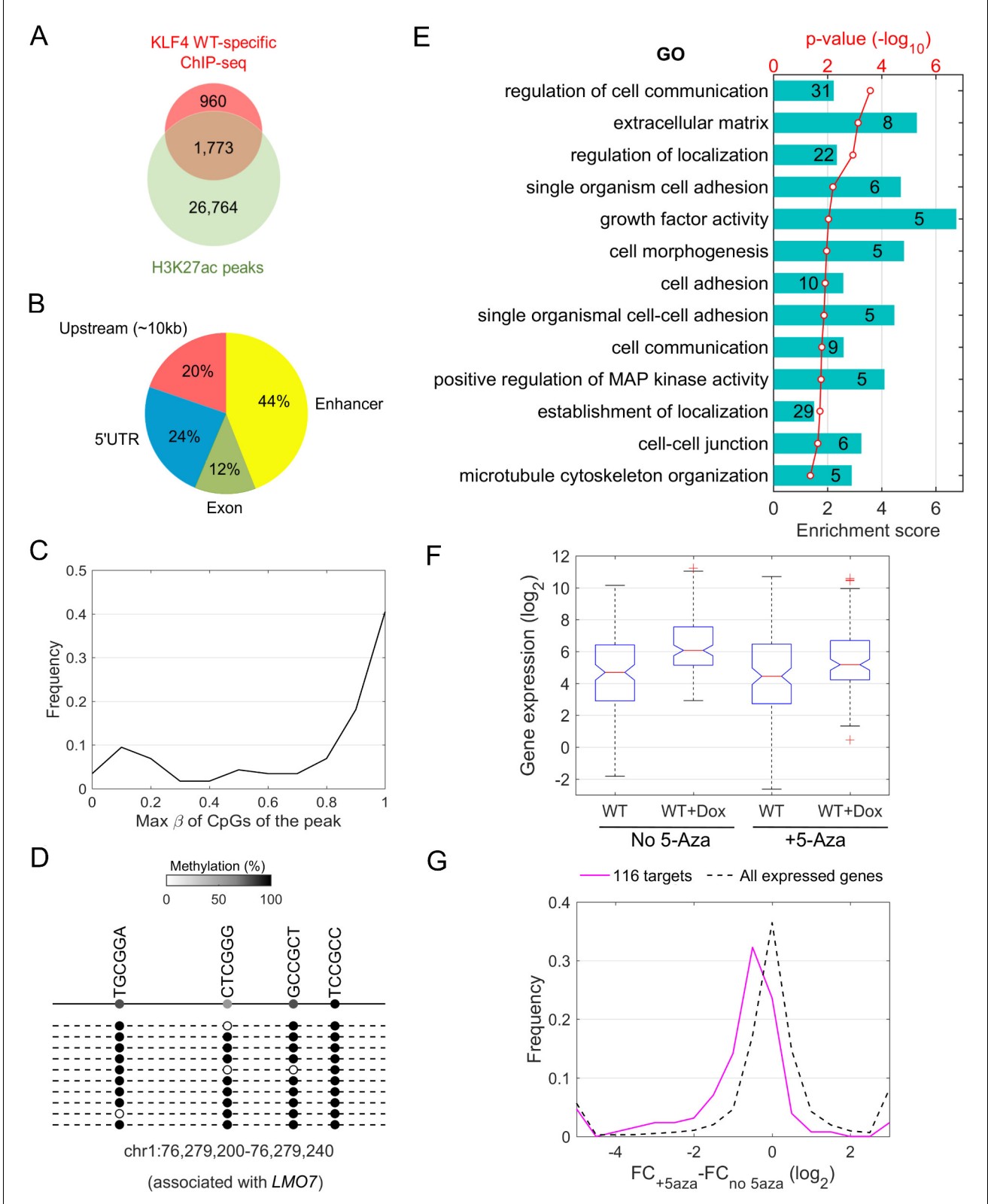

**Figure 4.** Downstream targets of KLF4-mCpG interactions. (**A**) Overlap between KLF4 WT-specific ChIP peaks (2733) and the enhancer mark H3K27ac ChIP-seq peaks. A total of 1733 loci were identified. (**B**) A total of 116 KLF4-mCpG direct targets were identified in a serial of genome-wide studies. The overlap between WT-specific binding peaks (2733) and the 308 WT-only upregulated genes indicated that 20%, 24% and 12% of these genes were activated by KLF4 binding to mCpGs in gene upstream, 5'UTR and exon region, respectively. The overlap between enhancer regions in KLF4 WT-

*Figure 4 continued on next page*

*Figure 4 continued*

binding sites further identified 44% genes were activated by KLF4 binding to mCpGs in the enhancer regions. (C) Methylation level distribution of cis-elements in KLF4 binding peaks associated with 116 target genes. (D) Bisulfite sequencing confirmed DNA methylation in some of the KLF4 WT-specific binding peaks. (E) Gene ontology analysis of direct targets of KLF4-mCpG indicated that these targets have been implicated in cell adhesion, migration, cytoskeleton arrangement and cell binding activities. (F) Boxplot of gene expression for WT, WT+Dox, WT + 5 Aza, and WT + 5-Aza+Dox (G) Histogram of FC (after and before Dox) difference with and without 5-Aza for 116 target genes and all expressed genes, respectively.

The following figure supplement is available for figure 4:

**Figure supplement 1.** Bisulfite sequencing of the cis-regulatory elements of additional KLF4-mCpG direct targets.

suggest that KLF4 could bind to methylated *cis*-regulatory elements to up-regulate crucial genes involved in tumor cell migration.

To examine the impact of global DNA demethylation on the expression level of KLF4-mCpG direct target genes, we performed whole genome RNA-seq in the presence of 5-Aza treatment. We observed that in the presence of 5-Aza, KLF4 WT induction did not activate 116 target genes significantly as that in the absence of 5-Aza (*Figure 4F*). Actually, the expression level of 101 of the 116 genes was reduced in 5-Aza treated cells as compared with untreated cells (*Figure 4G*), while about half (53%) of all expressed genes were down-regulated after 5-Aza treatment (p=1.2e-14), suggesting that DNA methylation is responsible for the transcription activation of these 101 genes targeted by KLF4-mCpG binding. These results further supported the conclusion that DNA methylation is directly involved in KLF4 binding and gene activation.

## KLF4-mCpG interactions activate *RHOC* via chromatin remodeling

To firmly establish the causality between mCpG-dependent KLF4 binding to *cis*-regulatory elements and transcription activation, we decided to focus on two important genes, namely *RHOC* and *RAC1*, for in-depth characterization. They encode two small GPTases that are known to play critical roles in cell migration and adhesion, and were found multiple times in the enriched GO terms as described above (*Karlsson et al., 2009*). For example, RHOC activation leads to F-actin stress fiber formation and the assembly of focal adhesion complexes. Studies reveal that there are no known *RHOC* pathogenic mutations found in cancers, however, biologically relevant aberrant levels of *RHOC* expression are common, and there is a strong association between *RHOC* expression levels and poor prognosis (*Karlsson et al., 2009*; *Narumiya et al., 2009*). An examination of the ChIP-seq datasets at the promoter regions of *RHOC* and *RAC1* indicated differential binding by KLF4 WT and R458A. Moreover, genome-wide bisulfite sequencing confirmed that these peaks contained highly methylated CpGs and these peaks were in close proximity to the coding regions of *RHOC* (−507 bp to TSS) and *RAC1* (−922 bp to TSS).

In the case of *RHOC*, we first showed that induction of KLF4 WT, but not the R458A mutant, could induce RHOC at both mRNA and protein levels using RT-PCR and immunoblot analysis, respectively (*Figure 5A,B*). Using a primer pair flanking this ChIP-seq peak, we confirmed that this fragment could be ChIP-ed substantially better with the KLF4 WT than the R458A mutant (*Figure 5C*). To further establish causality of DNA methylation-dependent binding of KLF4 on induction of its target genes, we performed bisulfite sequencing against this *cis*-regulatory element with and without the 5-Aza treatment. We observed that four of the five CpG in this region were 100% methylated; however, after the 5-Aza treatment the methylation levels of the four CpGs was reduced to various levels, ranging from 13–80% (*Figure 5D*). Most importantly, reduction of methylation almost completely abolished KLF4 WT binding to this *cis*-regulatory element (*Figure 5E*), and RHOC protein level was also significantly repressed (*Figure 5F*). The same set of assays was also carried out to analyze the *cis*-regulatory elements of *RAC1* and the same results were obtained (*Figure 5—figure supplement 1*). Therefore, high methylation levels in *RHOC* and *RAC1* promoters were essential for KLF4 WT to activate their gene transcription.

Because DNA methylation is usually considered to be associated with repressive histone marks to maintain downstream gene silencing, our observation suggested that the recruitment of KLF4 WT to the methylated *RHOC* promoter altered chromatin status in order to activate *RHOC* transcription. Based on KLF4's interaction with histone modifying enzymes, we examined the status of an active

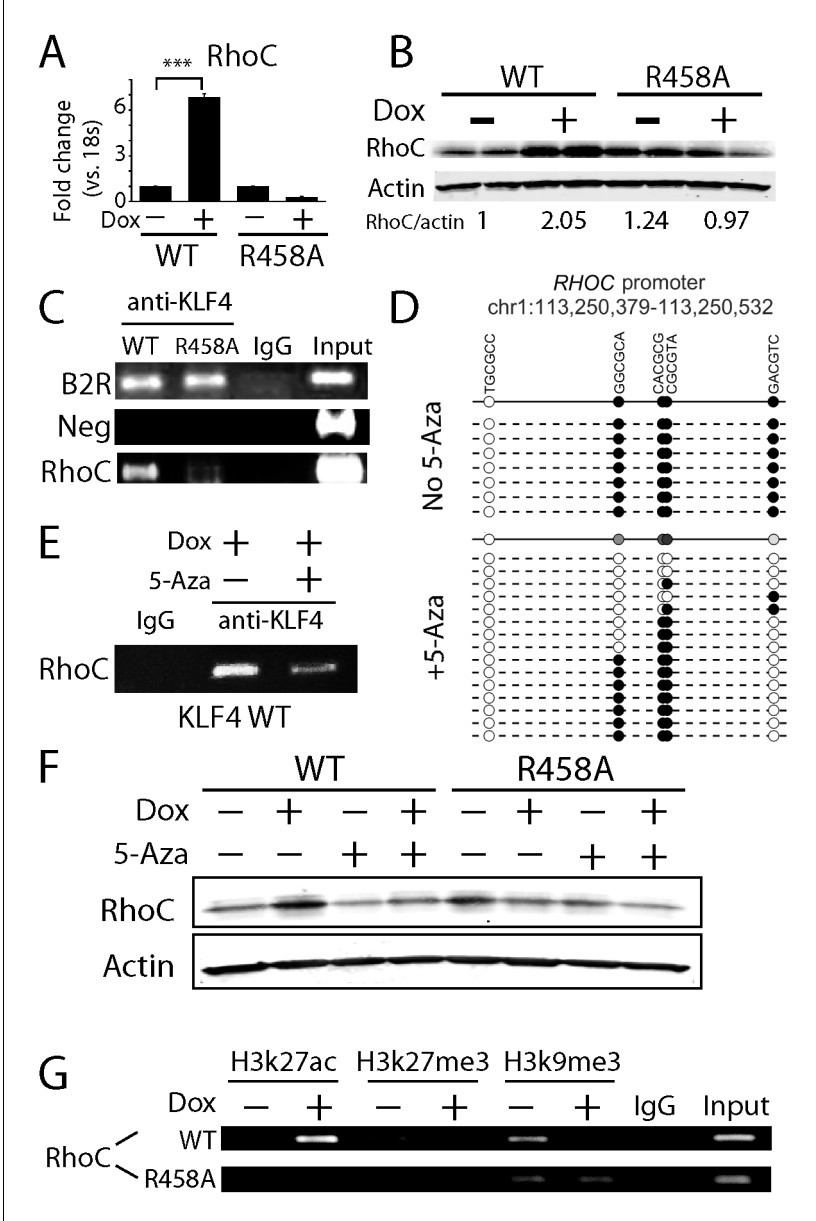

**Figure 5.** KLF4-mCpG binding activity activates RHOC expression by chromatin remodeling. (**A**) Real-time PCR indicated that *RHOC* RNA was significantly upregulated by KLF4 WT but not R458A (+Dox, 48 hr, ***p<0.001). (**B**) RHOC protein expression was upregulated by KLF4 WT only; fold changes were listed under the blots. (**C**) A KLF4 antibody was used to precipitate crosslinked genomic DNA from cells expressing KLF4 WT and R458A. Rabbit IgG was used to control for non-specific binding. De-crosslinked DNA samples were served as the input for PCR. The *RHOC* promoter was only enriched in the ChIP'ed samples from KLF4 WT expressing cells. In contrast, a known KLF4 binding site on B2R promoter was used as a positive control, and detected in the ChIP'ed samples from both KLF4 WT and R458A expressing cells. A non-promoter sequence was selected as a negative control (Neg) and no band was detected. (**D**) Bisulfite sequencing of *RHOC* promoter region: each row represents one sequenced clone; each column represents one CpG site; filled circles stand for methylation. All 7 clones showed 100% methylation at four CpG sites in the *RHOC* promoter (~507 bp upstream from TSS). The DNMT inhibitor 5-Aza pretreatment partially reversed DNA methylation in the *RHOC* promoter region. (**E**) ChIP-PCR indicated that 5-Aza greatly abolished KLF4 WT binding to *RHOC* promoter region when hypomethylated. (**F**) Immunobloting analysis showed that KLF4 WT-induced RHOC up-regulation was blocked by pretreatment with 5-Aza. (**G**) KLF4-mCpG interactions in *RHOC* promoter region triggered histone mark changes. Tet-on KLF4 WT cells were treated with Dox for 48 hr, and genomic DNA from cells before and after Dox treatment were ChIP'ed with antibodies recognizing different histone marks and amplified for the RHOC promoter region. KLF4-mCpG interactions were

*Figure 5 continued on next page*

*Figure 5 continued*

associated with an increase in the active mark H3K27ac and a decreased in the repressive marks H3K27me3 and H3K9me3. Disrupted KLF4-mCpG interactions in KLF4 R458A expressing cells did not show changes in these histone marks in the *RHOC* promoter region.

The following figure supplement is available for figure 5:

**Figure supplement 1.** Methylated DNA in the *cis*-regulatory region of *RAC1* determined KLF4 binding and *RAC1* upregulation.

(H3K27ac) and two repressive (H3K27me3 and H3K9me3) histone marks at the *RHOC* promoter regions. We also monitored histone mark changes before (0 hr) and after (48 hr) KLF4 induction. We observed that upon KLF4 binding to DNA (48 hr post Dox), the active mark H3K27ac was significantly increased by ~2.5 fold. On the other hand, both repressive marks, H3K9me3 and H3K27me3, were decreased by ~3.5 and~6 fold, respectively (*Figure 5G*). As a comparison, KLF4 R458A induction failed to trigger any detectable changes in H3K27ac, H3K9me3, or H3K27me3 marks in the *RHOC* promoter region (*Figure 5G*). Thus, the recruitment of KLF4 WT to the methylated *RHOC* promoter initiated conversion from repressive to active chromatins, as a prerequisite for activation of *RHOC* transcription.

## KLF4-mCpG interactions globally affect chromatin status

To determine whether KLF4 WT activated its other direct targets via chromatin remodeling as described above, we performed ChIP-seq analysis against one active (H3K27ac) and two repressive (H3K9me3 and H3K27me3) histone marks before (0 hr) and after (48 hr) KLF4 WT induction. For all of the three marks, we observed remarkable dynamic changes, centered around the 162 KLF4 WT ChIP-seq peaks associated with the 116 direct targets of KLF4-mCpG (*Figure 6A*). Specifically, H3K27ac level increased in 83.3% of the 162 peaks at 48 hr (*Figure 6B*, p=7.2E-19). In contrast, H3K27me3 and H3K9me3 levels decreased in 54.3% (p=3.4E-2) and 63.6% (p=1.5E-4) of the 162 peaks at 48 hr, respectively (*Figure 6B*). As a negative control, we examined the H3K27ac dynamic changes in the up-regulated genes shared by KLF4 WT and R458A. Only 36% of the shared KLF4 peaks (22/60) had stronger H3K27ac signal after 48 hr of Dox treatment (*Figure 6—figure supplement 1A–C*), suggesting that KLF4-mCpG binding enhances H3K27ac.

To further confirm that mCpG-mediated KLF4 binding caused chromatin remodeling on a genome-wide scale, we compared the H3K27ac ChIP-seq data obtained before (0 hr) and after (48 hr) KLF4 WT and R458A induction. An example was shown in *Figure 6C*. After KLF4 WT was recruited to intron region of LMO7, which is known to play a role in cell migration and adhesion (*Hu et al., 2011*), the H3K27ac level was significantly increased around the KLF4 ChIP-seq peak. In contrast, R458A mutation abolished the KLF4 binding at the same locus. As a result, the H3K27ac level remained low after R458A induction.

Overall, in KLF4 WT-induced cells, 3593 novel H3K27ac peaks were generated, while only a negligible number of H3K27ac peaks (41) disappeared at 48 hr of KLF4 WT induction (*Figure 6D*). Among the 3593 novel H3K27ac peaks, 274 were found in the flanking regions of KLF4 ChIP-seq peaks (p=3.5E-5). In contrast, induction of R458A did not yield noticeable changes in H3K27ac peak numbers. Taken together, these results suggest that mCpG-mediated KLF4 binding events induced the changes of histone modification and gene activation.

## Discussion

Abnormal DNA methylation has been found in many cancer types including glioblastoma. Contrary to traditional view of DNA methylation in gene silencing, recent survey of methylome and gene expression profiling demonstrated that the promoters of as many as 20% highly expressed genes in tumors are methylated at CpG sites, suggesting that at least in some cases, DNA methylation positively correlates with gene expression (*Feinberg, 2014*, *2007*). Although previous studies suggested that many TFs could interact with methylated DNA in vitro, it remains elusive whether such mCpG-dependent binding activity plays any significant physiological role in cells. To address this question,

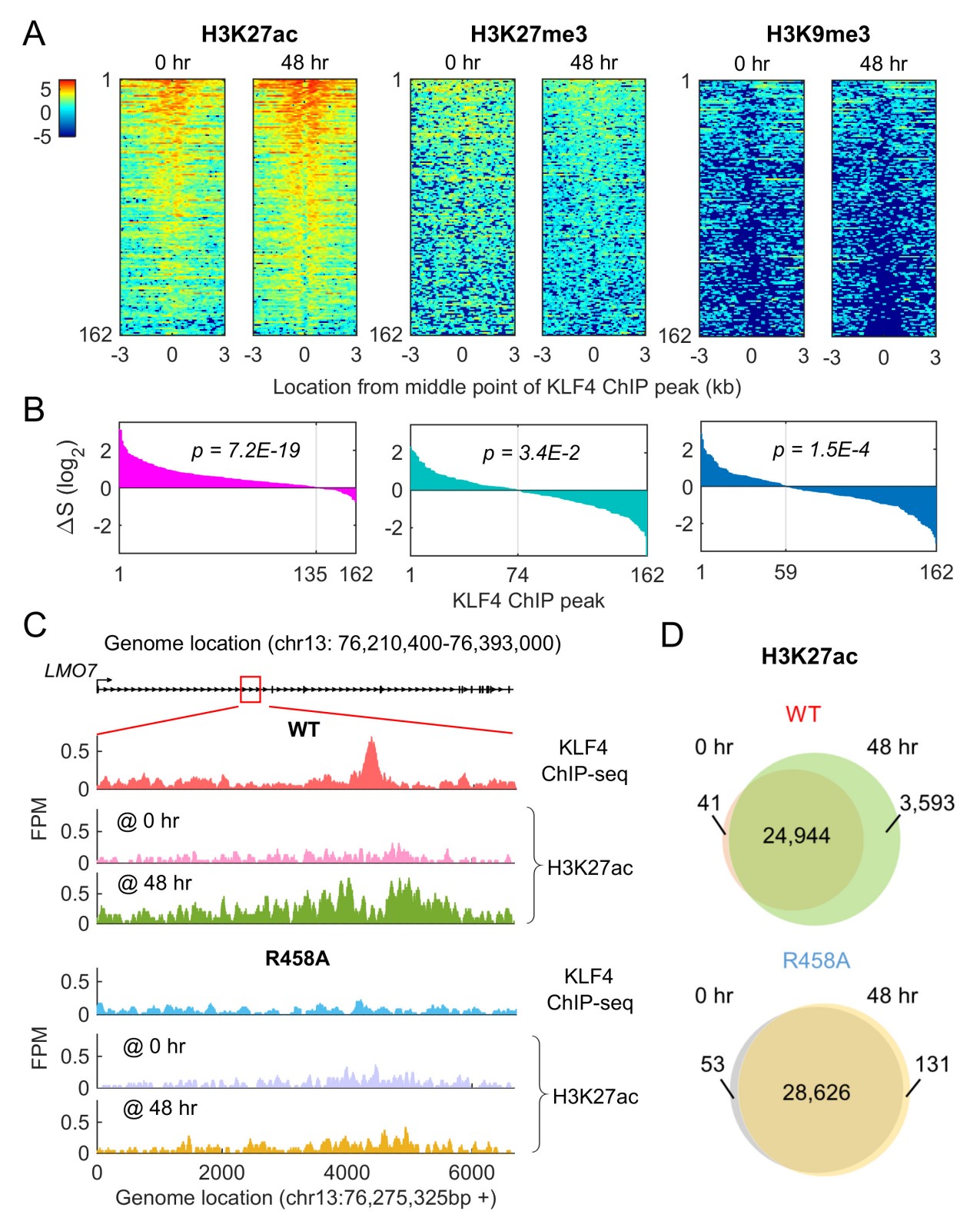

**Figure 6.** Methyl-mCpG dependent KLF4 binding activity triggers chromatin remodeling to activate gene expression. (A) Heatmaps of histone mark signals, H3K27ac, H3K27me3, and H3K9me3, before and after KLF4-mCpG interactions (0 vs. 48 hr), respectively, ±3 kb surrounding 162 KLF4 ChIP peaks, which were associated with the 116 KLF4-mCpG direct targets. The peaks were sorted by their average signals at 48 hr for each histone mark. (B) The signal difference of histone marks between 48 hr and 0 hr, sorted from minimum to maximum. Over 83% of the 162 peaks had increased H3K27ac

*Figure 6 continued on next page*

*Figure 6 continued*

signals (p=7.2E-19), whereas 54.3% (p=3.4E-2) and 63.6% (p=1.5E-4) of the peaks had decrease in H3K27me3 and H3K9me3 signals, respectively. (C) Stronger H3K27ac signals were accumulated surrounding the KLF4 WT ChIP-seq peak on gene *LMO7*, at 48 hr after KLF4 WT induction; whereas no significant change in H3K27ac signals was observed in R458A expressing cells, as the R458A mutation abolished KLF4-mCpG binding ability. (D) Genome-wide analysis of dynamic changes of H3K27ac peaks before and after KLF4 WT or R458A induction, respectively. A total of 3593 new H3K27ac peaks appeared after KLF4 WT induction (upper panel), whereas only 131 new peaks were generated in KLF4 R458A expressing cells (lower panel), indicating that mCpG-dependent KLF4 binding activity caused chromatin remodeling to activate gene expression.

The following figure supplement is available for figure 6:

**Figure supplement 1.** Dynamic changes of histone mark H3K27ac signal in the shared KLF4 peaks.

we took advantage of the R458A mutation that abolished KLF4's mCpG-dependent binding activity to dissect the functional impact of TF-mCpG interactions. We demonstrated that several strong phenotypes, ranging from cell morphology to cell adhesion and migration, were dependent on this newly discovered activity of KLF4. Using a series of in vivo assays coupled with bioinformatics analyses, we further showed that KLF4 could gain access to the inactive chromatin regions via binding to methylated DNA motifs and consequently, led to conversion from repressive to active histone marks and much enhanced transcription of the direct gene targets of KLF4 (*Figure 7*). Thus, we are the first to provide genome-wide evidence that recruitment of KLF4 to methylated *cis*-regulatory elements resulted in a global chromatin remodeling (i.e., from repressive to active states), which contradicts the classic view that CpG methylation is a result of chromatin remodeling (i.e., from active to repressive states).

Whereas many proteins have been found to recognize methylated DNA, the causality between DNA methylation and TF binding is not always straightforward. It has been reported that TF binding could change methylation status of *cis*-regulatory elements (*Feldmann et al., 2013*), which suggests that KLF4-mCpG binding could be the secondary effect of gene transactivation. To rule out the possibility that KLF4 binding results in methylation of its targeted genes, we performed bisulfite sequencing in tet-on KLF4 WT cells before and after KLF4 induction (0 hr and + Dox 48 hr). Unchanged methylation level in KLF4 binding site (e.g. *RHOC* promoter region) indicated that methylation is not a consequence of KLF4 binding (data not shown). Other forms of cytosine modification, including 5-hydroxymethyl-cytosine (5-hmC), 5-formylcytosine, and/or 5-carboxylcytosine also exist in cells (*Tan and Shi, 2012*). This will not affect our conclusion appreciably since KLF4 does not bind to 5-hmC (*Spruijt et al., 2013*).

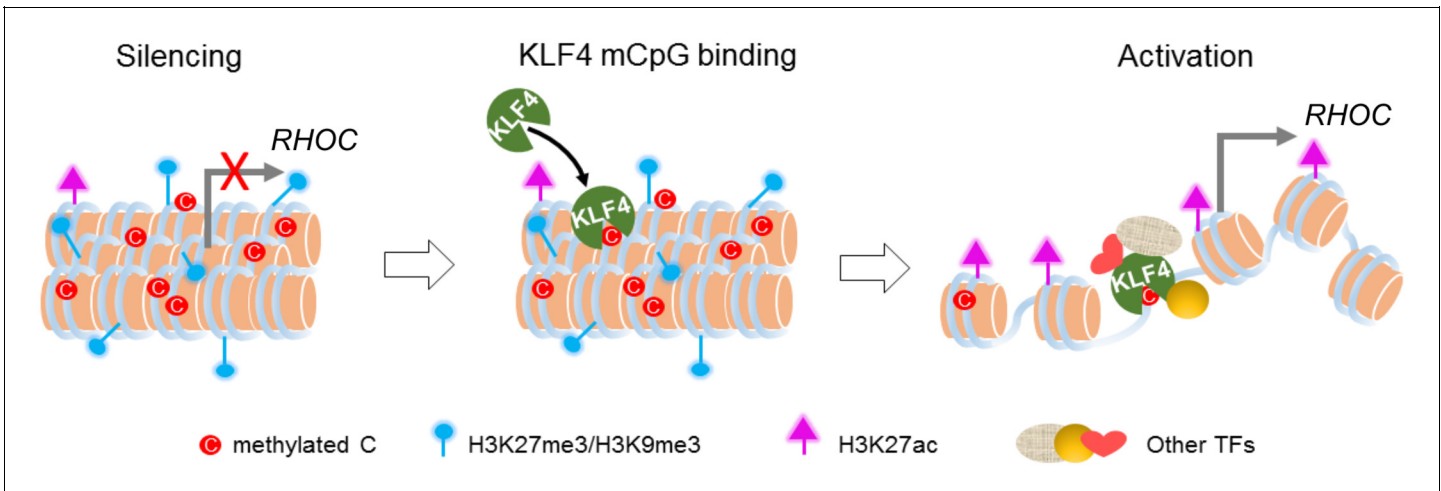

**Figure 7.** Working model of KLF4 binds to methylated *cis*-regulatory elements, followed by chromatin remodeling and transcription activation.

We are fully aware that 5-Aza is only a sub-optimal approach to examine the effect of specific methylation sites, because it induces global demethylation and may have some non-specific effects (*Christman, 2002*). However, we have multiple lines of evidences showing that the observed phenotypes are due to specific interactions between KLF4 and methylated sites. (1) In our control experiments when KLF4 R458A cells were treated with 5-Aza, we did not see changes in gene expression and cell migration. (2) In the focused studies of *RHOC* and *RAC1*, we showed that DNA methylation is responsible for KLF4 binding and gene activation. *RHOC* and *RAC1* are major GTPase involved in cell migration and motility. (3) Comparison of ChIP-seq between KLF4 WT and R458A showed that WT-specific binding sites are enriched for highly methylated sites. Given the phenotypic differences between KLF4 WT and R458A, the result suggested that the observed phenotypes in WT are likely due to specific interactions between KLF4 and DNA methylation sites. (4) Our new RNA-seq data in the presence of 5-Aza also indicated that 5-Aza globally reversed the up-regulation of many KLF4-mCpG targets.

In our study, we placed a higher priority on KLF4-mCpG-mediated gene transactivating effect, as this will be most paradigm-shifting based on current understandings that CpG methylation results in gene silencing. Here we found that KLF4 recognizes mCpG at proximal and enhancer regions to activate gene expression. Many questions remain regarding how DNA methylation, a repressive epigenetic mark, activates gene expression. In cells, opening the chromatin structure is often the first step for gene transcription (*Cirillo et al., 2002*; *Lupien et al., 2008*). KLF4 is one of the pioneer TFs in somatic reprogramming that interact with condensed chromatin and recruit histone-modifying enzymes to enable gene expression (*Buganim et al., 2013*; *Iwafuchi-Doi and Zaret, 2014*). Yet, exactly how these pioneer TFs gain access to condensed, highly methylated DNA is still unknown (*Drouin, 2014*). Although recent studies showed that pioneer transcription factors, including KLF4, are capable of recognizing partial motifs located in the heterochromatins, our study, for the first time, demonstrated that the pioneer transcription factor KLF4 could actually bind to methylated motifs located in repressive chromatins, providing an alternative mechanism for pioneer factors interacting with repressive chromatin. Our data support a new model in which TF-mCpG binding communicates with histone modifications to initiate gene transcription. The cross-talk between DNA methylation and histone modifications in gene regulation will open up a new avenue for determining how epigenetic mechanisms drive physiology and pathophysiology including tumor malignancy.

In summary, TF binding to methylated regions of the genome to activate gene transcription is a new paradigm supported by several studies (*Filion et al., 2006*; *Mann et al., 2013*; *Rishi et al., 2010*; *Sasai et al., 2010*; *Serra et al., 2014*; *Spruijt et al., 2013*). Our work is the first to demonstrate such gene activation mechanism can mediate physiological functions in biologically relevant systems (*Zhu et al., 2016*). We further demonstrated that mCpG dictates TF binding, histone modifications and gene activation in a sequence-specific manner, thereby influencing cancer cell phenotypes. Our study reveals that mCpG-dependent binding activity of KLF4 serves as a link between methylated CpG and changes in chromatin status (*Charlet et al., 2016*), two most important epigenetic mechanisms in gene regulation. In all, our study provides a new paradigm in which TFs can act as a new class of DNA methylation readers/effectors that drive gene transactivation and diverse biological processes.

## Materials and methods

### Cell culture and reagents

All reagents were purchased from Sigma Chemical Co. (St. Louis, MO) unless otherwise stated. The human glioblastoma (GBM) cell lines U87 (CLS Cat# 300367/p658_U-87_MG, RRID:CVCL_0022) and U373 (ATCC Cat# HTB-17, RRID:CVCL_2219) were originally purchased from ATCC (Manassas, VA) and cultured in our laboratory. Both cell lines are free from mycoplasma and authenticated with short tandem repeat (STR) profiling by Johns Hopkins Genetic Resources Core facility using Promega GenePrint 10 system (Madison, WI). U87 cells were cultured in Minimum Essential Media (MEM, Thermo Fisher Scientific, Grand Island, NY) supplemented with sodium pyruvate (1%), sodium bicarbonate (2%), non-essential amino acid (1%) and 10% fetal calf serum (FCS, Gemini Bio-products, West Sacramento, CA). U373 cells were cultured in Dulbecco's Modified eagle medium (DMEM, Thermo Fisher Scientific) supplemented with 2% (4-(2-hydroxyethyl)−1-piperazineethanesulfonic acid

(HEPES) and 10% FCS. Cells were incubated in a humidified incubator containing 5% $CO_2$/95% air at 37°C, and passaged every 4–5 days.

## Engineering tet-on GBM cells expressing KLF4 WT and KLF4 R458A

KLF4 WT and R458A constructs were inserted into a doxycycline-inducible TripZ lentiviral vector (Thermo Fisher Scientific) (*Ying et al., 2011a*). Virus was packaged using the Viral Power Packaging system (Thermo Fisher Scientific) according to the manufacturer's forward transfection instructions. Virus were collected by centrifuging at 3000 rpm for 15 min. GBM stable cell lines were established by transfecting the cells with lentivirus harboring tet-on KLF4 WT or KLF4 R458A constructs and selected with puromycin (1 µg/ml). The introduction of doxycycline (Dox) to the system initiates the transcription of the genetic product. Cells without doxycycline treatment serve as controls for KLF4 function.

## Immunoblot and immunocytochemistry

Immunoblot analysis was used to examine KLF4 protein expression. To collect whole cell protein, cells were lysed with RIPA buffer (50 mM Tris-HCl, pH 7.4, 150 mM NaCl, 1% NP-40, 0.25% Na-deoxycholate) containing protease and phosphatase inhibitors (EMD Millipore, Billerica, MA) and sonicated for 15 s; the suspensions were centrifuged at 3000 *g* for 10 min. Thirty micrograms of protein were separated using 4–20% SDS-PAGE gels (Lonza, Williamsport, PA) and blotted onto nitrocellulose membranes (*Reznik et al., 2008*). Membranes were incubated in Odyssey Licor blocking buffer (LI-COR Biosciences, Lincoln, NE) for 1 hr at room temperature and then overnight with primary antibodies at 4°C in Odyssey blocking buffer. After rinsing, membranes were incubated with IRDye secondary antibodies (1:15000-1:20,000, LI-COR Biosciences) and protein expression changes were quantified by dual wavelength immunofluorescence imaging (Odyssey Infrared Imaging System, LI-COR Biosciences) as previously described (*Ying et al., 2011b*). Antibodies were purchased from: anti-KLF4 (Santa-Cruz, Dallas, Texas); anti-RHOC (Cell signaling, Danvers, MA); and anti-RAC1 (Cell signaling).

For staining, GBM cells grown on glass slides were fixed with 4% paraformaldehyde for 30 min at 4°C and permeabilized with PBS containing 0.1% Triton X-100 for 10 min. The cells were then incubated with primary antibodies at 4°C overnight and then incubated with appropriate corresponding secondary antibodies conjugated with Alex Flourescent 488 or cy3 for 30 min at room temperature. For double staining with F-actin, Alex Flourecent 647-conjugated phalloindin (1 unit, Thermo Fisher Scientific) was used to incubate with the cells. Slides were mounted with Vectashield Antifade solution containing DAPI (Vector Laboratories, Burlingame, CA) and observed under fluorescent microscopy. Immunofluorescent images were taken and analyzed using Axiovision software (Zeiss, Germany).

## Cell adhesion and migration assay

For adhesion assay, twenty-four well plates were blocked with Dulbecco's Modified Eagle *Medium* (DMEM) containing 0.5% bovine serum albumin (BSA) for 1 hr at 37°C followed by plating GBM cells at a density of $4 \times 10^5$ cells/well. After 1 hr incubation at 37°C, plates were shaken at 2000 rpm for 15 s, washed twice with pre-warmed DMEM medium containing 0.1% BSA, and the number of remaining adherent cells were measured with MTT assays (*Xia et al., 2005*).

For migration assays in transwells (Corning, Lowell, MA), GBM cells were suspended at $1 \times 10^6$ cells/ml; 100 microliters of the cell suspension were added to the upper chamber of the transwells in serum-free medium (*Wang et al., 2012*). Six hundred microliters of medium containing 10% FCS was added to the lower chamber. After 3 hr incubation at 37°C, cells were fixed with Diff-Quick kit (Thermo Fisher Scientific). Cells on the upper side of the transwells were gently wiped off with Q-tips. Cells migrating through the filter were stained with 4'−6-Diamidino-2-phenylindole (DAPI). Migration was quantified by counting cells on five selected fields of view per transwell in at least three independent experiments (*Wang et al., 2012*).

For wound healing assays, GBM cells were grown under 10% FCS medium in 35 mm dishes until confluent. Cell proliferation was inhibited by mitomycin C (1 µg/ml) for half hour. Several scratches were created using a 10 µl pipette tip through the confluent cells. Dishes were washed with PBS for three times and cells were grown in 0.1% FCS medium for 24–48 hr. Phase contrast pictures were

taken at different time points. The width of the scratch was measured and quantified as previously described (*Goodwin et al., 2010*).

## Reverse-transcriptase PCR and quantitative real-time PCR

Total RNA (1 µg) was reverse-transcribed using the oligo (dT)12–18 primer and Superscript II (Thermo Fisher Scientific) according to the manufacturer's instructions. The thermal cycling conditions were as follows: 95°C, 5 min, followed by 30 cycles of 95°C for 10 s, 55°C for 10 s, and ended with 72°C for 30 s.

Data were analyzed using parametric statistics with one-way ANOVA. Post-hoc tests included the Students T-Test and the Tukey multiple comparison tests as appropriate using Prism (GraphPad, San Diego, CA). All experiments reported here represent at least three independent replications. All data are represented as mean value ± standard error of mean (S.E.) significance was set at $p < 0.05$.

## Genome-wide profiling of gene expression and KLF4 binding in GBM cells

### RNA-seq

RNAs from KLF4 WT and R458A-expressing cells (0 hr and 48 hr +Dox) was subjected to Illumina HiSeq next generation sequencing following the standard amplification and library construction protocol provided by the Johns Hopkins Deep Sequencing and Microarray Core Facility. Sequencing was performed using 76-base single-end reads, with 23- to 33 million reads generated from each sample. We first used Tophat2 to map all reads to human genome (hg19) then employed Cufflink to summarize the gene/transcript expression based on mapped reads. An R package, DEGseq, was taken to identify DEGs for $p < 0.001$ between 0 hr and 48 hr in KLF4 WT and R458A cells, respectively.

## Whole genome bisulfite sequencing (WGBS) analysis

We employed the software package bismark (*Krueger and Andrews, 2011*) to perform WGBS analysis. First we built bismark reference human genome, then mapped sequence reads onto these specific references. Two files were generated afterwards. The text file includes the summary about total reads, mapping efficiency, total methylated C's in CpG/CHG/CHH context. The other file in the same format was used for next step to extract methylation. Finally, we used bismark2bedGraph followed by coverage2cytosine to achieve the methylated and unmethylated reads of all CpG sites. The $\beta$ value was calculated for each CpG site as the ratio of number of methylated reads to sum over methylated and unmethylated.

### Motifs analysis

To identify methylated motifs enriched in KLF4 WT-specific peaks, we first used WGBS information to selected all 6-mers including mCpG ($\beta > 0.6$), then enumerated these 6-mers to compare their occurrence in 2733 KLF4 WT-specific peaks and all KLF4 binding peaks. The p-values were calculated based on hypergeometric model to represent the significance of methylated 6-mers' frequencies in KLF4 WT-specific peaks compared to all, followed by multiple-test Bonferroni correction. The 6-mers with $p < 0.01$ were selected to construct the motif logo. The package, MEME (Multiple EM for Motif Elicitation) (*Bailey and Elkan, 1994*), was used to evaluate motifs significantly over-represented in KLF4 shared peaks, compared to all KLF4 binding peaks.

### Gene ontology analysis

Gene Ontology analysis (*Ashburner et al., 2000*) was performed for the 116 differential expressed genes (DEGs) up-regulated by WT only, compared to that for total 12,824 genome-wide expressed genes (FPKM >0.5). The statistical significance of the enrichment was evaluated by p-value based on hypergeometric distribution model. The p-values were then adjusted by multiple-test correction via false discovery rate (FDR). A cutoff of FDR < 0.05 was used to identify significantly enriched GO terms.

## Chromatin immunoprecipitation (ChIP)-seq

A commercial ChIP-grade anti-KLF4 antibody (H180; Santa Cruz) recognizing the N-terminal region of KLF4 (DNA-binding domains of KLF4 are located to the very C-terminus) was used for ChIP. Tet-on KLF4 WT and R458A cells were treated with Dox for 48 hr followed by ChIP using the anti-KLF4 antibody and Dynabeads Protein A/G (Thermo Fisher Scientific) according to a protocol described previously (*Hu et al., 2013*). DNA library construction and sequencing was performed at Johns Hopkins Deep Sequencing and Microarray Core Facility. The antibodies used for ChIP experiments were as follows: anti-KLF4 (Santa Cruz, H-180, sc-20691); anti-H3K27ac (Abcam, ab4279) (*Hawkins et al., 2010*); anti-H3K27me3 (Millipore, 07–449) (*Hawkins et al., 2010*) and anti-H3K9me3 (Abcam, ab8898) (*Hawkins et al., 2010*).

KLF4 ChIP-Seq data were mapped by Bowtie2, followed by MACS 1.4 being used to call peaks with cutoff of p<1E-5. We first obtained binding peaks for KLF4 WT and R458A, respectively. The peaks identified for both KLF4 WT and R458A at the same locus were referred as shared KLF4 binding peaks. Then we used KLF4 WT as foreground and R458A as background control to call peaks again. The new peaks were marked as KLF4 WT-specific ChIP peaks, only if they were not overlapped with shared ones which had been already identified. Same approach was used to obtain R458A specific binding peaks for which the foreground was KLF4 R458A ChIP-Seq data compared to the background of KLF4 WT ChIP-seq data.

We utilized MACS2 to recognize broad peaks of H3K27ac based on their ChIP-Seq data mapped by Bowtie2. The cutoff of broad peak call was q < 0.1. The same procedure as that for KLF4 ChIP-Seq was taken to distinguish H3K27ac peaks at 0 hr only, at 48 hr only, or shared at both 0 hr and 48 hr, for KLF4 WT and R458A, respectively.

## Quantitative study of histone modification

We chose two sets of genomic regions to quantitatively study dynamic changes of histone modification between 48 hr and 0 hr. The first group is regions within ±1 kb from the middle points of KLF4 binding peaks. As negative control, we selected other 7780 regions, which locate 4 kb to 6 kb away from the boundaries of 3890 KLF4 binding peaks, then removed 558 out of 7780 regions in the negative control group which overlap with at least one of KLF4 binding peaks. The signal of the histone modification within the region was represented by RPM (Reads per Million) in logarithmic scale base 2. The difference of signals between 48 hr and 0 hr was further normalized so that the difference for 7222 random regions was 50% up and 50% down.

All the raw data for our large-scale studies have been deposited in GEO (GSE97632).

## Assessment of CpG methylation status by bisulfite sequencing

Sanger bisulfite sequencing was performed as previously described (*Hu et al., 2013*). Purified genomic DNA from GBM cells were treated by EZ DNA Methylation-Direct Kit (Zymo Research, Irvine, CA). After bisulfite conversion, regions of interest were PCR-amplified using Taq polemerase. The primers used for bisulfite sequencing were listed in *Supplementary file 2*. PCR products were gel-purified and cloned into a TA vector (Thermo Fisher Scientific). Individual clones were sequenced (Genewiz, Cambridge, MA) and aligned with the reference sequence.

## Acknowledgements

We thank Dr. J Laterra for critical reading and H Lopez-Bertoni for technical expertise. This work was supported by grants from NIH R01NS091165 (SX), EY024580 (JQ), EY023188 (JQ), GM111514 (HZ and JQ), R01 GM111514 (HZ), R33CA186790 (HZ), U54 HG006434 (HZ), U24 CA160036 (HZ), P01NS097206 (HS) and R35NS097370 (GM).

## Additional information

### Funding

| Funder | Grant reference number | Author |
| --- | --- | --- |
| National Institutes of Health | R01NS091165 | Shuli Xia |

| National Institutes of Health | EY024580 | Jiang Qian |
| National Institutes of Health | R01 GM111514 | Heng Zhu |

The funders had no role in study design, data collection and interpretation, or the decision to submit the work for publication.

## Author contributions

JW, Conceptualization, Data curation, Software, Methodology, Writing—original draft, Writing—review and editing; YS, QS, BT, OO, SL, Data curation; MY, Resources; G-lM, Funding acquisition; HS, Resources, Funding acquisition; JQ, Conceptualization, Formal analysis, Supervision, Funding acquisition, Methodology, Writing—original draft, Writing—review and editing; HZ, Conceptualization, Formal analysis, Supervision, Funding acquisition, Writing—original draft, Writing—review and editing; SX, Conceptualization, Data curation, Formal analysis, Supervision, Funding acquisition, Validation, Investigation, Methodology, Writing—original draft, Project administration, Writing—review and editing

## Author ORCIDs

Olutobi Oyinlade, http://orcid.org/0000-0002-8932-2764
Shuli Xia, http://orcid.org/0000-0001-5849-6967

# Additional files

## Supplementary files

• Supplementary file 1. Direct downstream targets of KLF4-mCpG binding activity in GBM cells.

• Supplementary file 2. Primer sequences used for bisulfite-PCR.

## Major datasets

The following dataset was generated:

| Author(s) | Year | Dataset title | Dataset URL | Database, license, and accessibility information |
| --- | --- | --- | --- | --- |
| Wan J, Su Y, Song Q, Tung B, Oyinlade O, Liu S, Ying M, Ming G, Song H, Qian J, Zhu H, Xia S | 2017 | Methylation DNA mediated KLF4 binding activity in glioblastoma cells | https://www.ncbi.nlm.nih.gov/geo/query/acc.cgi?acc=GSE97632 | Publicly available at the NCBI Gene Expression Omnibus (accession no: GSE97632) |

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
