## [Decision Letter]

Thank you for submitting your article "Methylated cis-regulatory elements mediate KLF4-dependent gene transactivation and cell migration" for consideration by *eLife*. Your article has been reviewed by three peer reviewers, one of whom, Bing Ren (Reviewer #1), is a member of our Board of Reviewing Editors, and the evaluation has been overseen by Kevin Struhl as the Senior Editor.

Your work has been considered by the editors and three reviewers. The comments from the reviewers are attached. The reviewers recognize the potential significance of your finding of the function of DNA methylation-dependent binding of Klf4 in cell adhesion and migration, but raised considerable concerns regarding experimental evidence and the conclusions. In light of these concerns we cannot recommend publication of this manuscript. However, if you are able to address fully the main concerns as listed below then we would be happy to consider a much-revised version of the manuscript.

Essential revisions:

1) New experimental evidence is required to fully establish the DNA methylation dependent binding of wild type and the mutant form. Control experiments with Klf4 mutant lacking DNA binding potential altogether needs to be conducted; DNA methylation and motif analysis of the cell systems need to be performed.

2) Causality of DNA methylation dependent binding of Klf4 on the genome and induction of some target genes needs to be more fully established. As pointed by reviewer #3, the presence of the binding within 10kb of the promoter does not prove direct targeting relationship. New experimental data should be provided to support the notion that a target gene is directly regulated by Klf4's binding to the methylated DNA near the gene. This could be CRISPR/cas9 mediated genome editing or other means.

3) The treatment of cells by 5-Aza is inducing global DNA demethylation, and the observed phenotypes in the cell system can be attributed to non-specific effects. The authors need to discuss this aspect of the experiments to avoid drawing un-substantiated conclusions.

4) Statistical methods need to be better described and experimental data need to be more rigorously interpreted. Please check their comments carefully and make sure to remove or modify the particular statements in the revised manuscript.

Reviewer #1

In this manuscript Wan et al. reported the finding that Klf4 can indeed bind methylated DNA to activate transcription of genes in mammalian cells. Work in the laboratory of one of the co-authors (Dr. Zhu) previously showed that Klf4 could recognize DNA motifs with methylated CpG, and a point mutation R458A disrupts its methyl-DNA binding but not affect the binding to unmethylated sequences. In the current work, the authors tried to demonstrate function of the methyl-CG binding of Klf4. They chose two gliobastoma multiforme (GBM) cell lines U87 and U373 expressing either an inducible Wild Type Klf4 or one with R458A mutation. They found that upon induction of WT Klf4, but not MT form, the GBM cells acquire cell adhesion and migratory properties. They further investigated the binding sites of the two forms of proteins, and identified a substantial number of binding sites for the WT form but not the methyl-binding defective mutant. The also carried out RNA-seq analyses to identify genes regulated differentially by the WT versus Mutant form, and linked some of them to the differential binding of the Klf4 proteins. They claimed to identify 116 genes activated by mCpG-dependent KLF4 binding activity. In order to link the differential binding to DNA methylation at DNA, the group picked 15 regions and performed bisulfite sequencing. They showed that 10 of them contain DNA methylation. The authors conclude that Klf4's methyl-CG binding activity is critical for its role in mediating cell adhesion and migration in the GBM cells, and its binding to the methylated sites results in chromatin remodeling and transcription activation.

The work represents an in-depth follow up study of a phenomenon that some transcription factors binds to methylated DNA. The authors provided multiple lines of evidence supporting the functional role of methyl-CpG binding of Klf4. However, they also made some over-statement, and some conclusions are not sufficiently established with existing evidence. Additional experiments and analyses are needed to strengthen these conclusions.

Major problems:

1) Differential binding of WT and R458A Klf4 in GBM cells: Is this due to failure of the mutant protein to bind methylated DNA? Unfortunately, the evidence presented is weak: the authors performed Bisulfite Sequencing on 15 sites and found that 10 of them contain methylated CpG. Are these DNA sequences recognized by WT Klf4 but not mutant klf4? EMSA should be performed to establish this point. Further, 15 is too small a number to generalize to the rest of the differential binding sites. It is important to carry out base-resolution analysis of DNA methylation in these cells and determined if differential DNA binding by WT and R458A Klf4 in the GBM cells is indeed correlated with presence of DNA methylation at the Klf4 binding motifs in these regions.

2) The authors claimed that 116 genes are activated by mCpG-dependent KLF4 binding activity. Again, the evidence to support this claim is weak. Besides the reasons outlined in #1, there is also a need to establish the causality of KLF4 binding to methylated DNA in the cells and activation of these genes. Since the authors observed partial loss of cell adhesion and migration after 5-AZA treatment, it is important to show that the induction of these 116 genes is affected by 5-AZA in the direction that is consistent with their being targeted by Klf4.

3) The 5-AZA treatment leads to partial loss of migration of cells expressing WT Klf4. Is this due to DNA methylation dependent binding of Klf4? It is necessary to demonstrate loss of DNA methylation and loss of Klf4 binding at some sites after treatment.

Reviewer #2:

Binding of transcription factors to methylated regions of the genome to activate gene transcription is a new paradigm, and several studies have provided convincing evidence supporting this hypothesis. The manuscript by Wan et al., 'Methylated cis-regulatory elements mediate KLF4-dependent gene transactivation and cell migration' supports this hypothesis. In this study Wan et al. provides the best evidence of transcription factors binding to methylated DNA motifs and mediating gene activation. The experiments support their hypothesis convincingly and the manuscript describes it well. However, there are certain aspects that need to be addressed before it can be accepted.

Previously, the authors discovered that KLF4 can bind to methylated motifs and that a mutation in KLF4, (R458A) inhibited binding to methylated motifs but not unmethylated motifs (Hu et al., *eLife* 2013). They used this mutation as a negative control in the examination of KLF4 binding to methylated DNA sequences. However, in the present manuscript, they do not show that KLF4 R458A is able to activate unmethylated motifs in their genomic assays as they showed previously in transfection experiments. They need a negative control with no KLF4 function.

The authors make stable cell lines in human glioblastoma U87cells. They make doxycycline inducible cells and do cellular and genomic experiments to probe the function of KLF4 binding to methylated DNA. As already stated, they need a third cell with no KLF4 function, it can be the parental cells.

Major points:

1) The authors observe KLF4 ChIP-seq in WT and R458A transfected cells. There are some key aspects that need to be included. How many / percentage of canonical and methylated sites are specific or shared between WT and R458A cells. A representation of the motifs needs to be addressed. Figure 3, the authors need to provide the input track to the screenshot; it seems to be the background is high. Some other similar screenshots of genes as supplementary figures will be useful. MACS 1.4 has been used to call peaks for KLF4 while again MACS2 has been used to call peaks while comparing with the acetyl mark. It would be better if the authors use the same pipeline to call peaks, as they are different. Why do the authors need to call a broad peak in this case?

2) The R458A mutant binds to ~1200 regions, the authors need to provide details regarding these regions.

3) The methods do not indicate whether biological replicates were performed for the sequencing based assays, in that case reproducibility is questionable. The authors can also provide a supplementary summary table for the mapped reads for all the sequencing experiments.

4) Figure 1: KLF4 expressed at a much higher level in the mutant version and can be observed consistently across the doses, still the authors find loss of physiological functions and KLF4 binding peaks needs explanation. The authors need to include a negative control or some KLF4 silencing assay to show WT KLF4 is directly correlated to the observed physiological changes such as cellular migration / wound healing / adhesion.

5) Figure 2: Comparatively lesser focal location of F-actin and Vaculin can also be noted in R458A cells it will be helpful to if the authors show fields with similar number of cells as that of the WT cells.

6) Figure 5 subsection “KLF4-mCpG interactions activate RHOC via chromatin remodelling” It is evident treatment of WT cells with 5-aza and doxycycline induces expression of RHOC, so it is apparent expression of RHOC is not absolutely methylation / KLF4 dependent. The authors should explain this in their text. Authors should have R458A version of the figure.

7) What is the methylation status of the WT specific KLF4 chip peaks sharing acetylation mark.

*Reviewer #3:*

The paper has a single major finding which is that the KLF4 R458 mutation, which presumably affects nothing other than binding to methylated DNA, is severely compromised in its ability to induce both cellular phenotypes and gene expression changes in GBM cell lines. Moreover, these changes induced by WT KLF4 are largely abrogated by removal of DNA methylation. If one assumes that R458 has no other role, and that methylation has no indirect impact on the effects of KLF4 binding, then the core findings do seem to be supported by several lines of evidence.

That said, the paper makes a series of claims that are not supported by the data, and contains interpretations that I believe are erroneous. It also seems to contain some redundancy. It strikes me that the paper would be stronger if most of the material presented was removed and only the essence was retained. I have grouped my major concerns into four points, some of which are applicable to multiple sections of the paper.

Major points:

1) The statistical analyses are incompletely described, interpreted incorrectly, confuse correlation with causality, and contain biases.

The RNA-seq, ChIP-seq, and bisulfite sections need to state which overlaps are significant, and what is the predictive power, otherwise there is a strong potential that the manuscript is mainly describing phenomena with little or no significant relationship, and/or is confusing weak correlations or random overlaps with causality, due to unwarranted conditioning on unproven expectation. This is relevant to several sections, starting with this one:

Subsection "Identify direct targets of mCpG-dependent KLF4 interactions in GBM cells" is flawed. 613 genes are identified as responsive to KLF4 overexpression (induced more than 2-fold). Then, "To determine which genes were directly activated by mCpG-dependent KLF4 binding events, ChIP-seq identifies 3,890 peaks. 65 of these bind within 10kb of the promoter, or within the UTRs or exons of the induced genes, "indicating that most likely they were direct targets of KLF4-mCpG interactions" (this is the key point, so I quote from the text). I disagree that this shows that they are direct targets – the null hypothesis should be that there is no relationship, and in fact if there is any overlap it seems to be low. It is entirely possible that there is a random association between the ChIP data and the RNA-seq data, and even if it is nonrandom, the fold enrichment over random should be given.

Instead, this section concludes "Taken together, a total of 116 genes were directly [my emphasis] up-regulated via mCpG-dependent KLF4 binding to the cis-regulatory elements in the proximal regulatory regions (56%) and distal enhancers (44%)". There is simply not the evidence in this paper to make that conclusion.

Since we don't have the right numbers in the current paper I will explain why the reasoning is flawed using typical numbers seen in other studies (which seem to be similar here). Let's say that there are 10,000 active genes, peaks are found in 900, 600 are up-regulated, and 60 overlap. So, peaks are found in 9% of all promoters, but 10% of up-regulated promoters. 6% of genes are up-regulated, and 7% of genes with a peak are up-regulated. Certainly there is something statistically significant happening, but the relationship between binding sites and expression responses is weak, so even when we see it, we cannot infer causality. We know that many binding sites do nothing (at least 93%), and that many genes are induced with no binding at all (at least 90%). So, it is certainly possible that there could be genes that have a binding site that does nothing and are induced indirectly.

By analogy, if the same numbers were used for a high-cholesterol diet and heart attacks, we would be wrong to assume that all 60 people with heart attacks and a high cholesterol diet were caused by the diet, since certainly the diet is not deterministic, and there are multiple causes of heart attacks. Instead, we can conclude that seven of the heart attacks can be attributed to the diet, but we can't tell which seven they were.

In addition, two technical modifications of these analyses are needed to further avoid bias. First, the comparisons among binding sites, induced genes, marks etc need to be restricted to genes that are expressed at all in these cells – it is possible to get high overlaps among unrelated regulators only because everything tends to pile up on open chromatin and active promoters. Similarly, the GO analyses later in the paper should be done taking the induced genes as a background. KLF4 overexpression induces migration and morphological changes, which could explain why genes related to this process are expressed. It is possible that KLF4 only directly induces a small number of genes, and the rest of what we are seeing is a secondary effect.

*eLife* asked me to write a short review, but the same issues pervade subsequent analyses of overlaps with histone ChIP data and methylation. It is abundantly clear from ENCODE that there are lots of pervasive marks of all kinds in all cell types, many of which concentrate in active regions. Not every overlap is functionally significant, however, and in fact it is likely that most of them are doing nothing at any given time.

2) It seems possible that global DNA demethylation could attenuate the ability of cells to do many things, and thus the results shown in Figure 2 could be nonspecific. Are there other perturbations that can trigger the same responses that are being assayed, and are these responses intact in aza-treated cells? If not, then the effects cannot really be attributed to a specific effect of KLF4 binding.

3) I do not see that Figure 5 adds anything to the paper. It is simply a reiteration of assays already used earlier on a global scale, but using PCR assays. The same issues I describe above in major points 1 and 2 also apply to these results – I believe that the final statement "Therefore, high methylation level in RHOC promoter is essential for KLF4 WT to activate RHOC transcription" is rather an overstatement. The first sentence is better supported: "Several lines of evidence supported that DNA methylation mediated RHOC activation by KLF4 WT". On the whole, however, this is the same argument that has already been made in genome-wide analyses, with no new evidence. I would suggest that this section is dispensible, since it only confirms what would be expected from the rest of the manuscript.

4) The association of active marks with induced promoters and with many of the Klf4 sites is unsurprising – this would be seen even if the induction of the supposed "target genes" was indirect, and the KLF4 binding was coincidental. Indeed, the fact that there are 3,593 novel H3K27ac peaks, but only 274 overlap KLF4, shows that the two are largely dissociated. Thus, the statement in the Discussion that "Our study reveals that mCpG-dependent binding activity of KLF4 serves as the link between methylated CpG and changes in chromatin status" is not supported by the data shown. It is "a" link, but not "the" link.

---

## [Author Response]

*Essential revisions:*

*1) New experimental evidence is required to fully establish the DNA methylation dependent binding of wild type and the mutant form. Control experiments with Klf4 mutant lacking DNA binding potential altogether needs to be conducted; DNA methylation and motif analysis of the cell systems need to be performed.*

To fully examine the DNA methylation status of KLF4 WT and R458A ChIP-seq peaks, we performed whole genome bisulfite sequencing to decode the methylome of U87 cells and combined the DNA methylome data separately with the KLF4 WT and R458A ChIP-seq datasets. We found that 66% of the KLF4 WT-specific ChIP-seq peaks showed a high methylation level (e.g., β > 60%) at CpG sites, while only 36% of the ChIP-seq peaks shared by KLF4 WT and R458A reached a similar CpG methylation level (*p* = 3.7e-223). Different cutoffs for defining high methylation levels did not alter this observation (Figure 3). Therefore, the KLF4 WT-specific ChIP peaks are enriched for highly methylated CpGs. In the revised manuscript, we have incorporated the new data in Figure 3.

Next, we carried out motif analysis to identify enriched and highly methylated 6-mer DNA motifs in WT-specific ChIP-seq peaks, as well as in shared ChIP-seq peaks. At a cutoff of β > 60% CpG methylation we found 10 methylated 6-mer motifs that were significantly over-represented in the WT-specific peaks (*p* = 6.6e-37). Many of them share sequence similarity to the motif 5’-CCCGCC (Figure 3, of which the methylated form was reported to be recognized by KLF4 in our previous study (Hu et al., *eLife* 2013). In contrast, the peaks shared by WT and R458A were found to be enriched for different motifs (e.g., 5’-AAAAGGAA and 5’- GAGTTGAA). The new data is now in Figure 3. Taken together, these new data confirmed that the KLF4 WT-specific ChIP-seq peaks were enriched for highly methylated KLF4 binding motifs.

We respectfully disagree that “control experiments with Klf4 mutant lacking DNA binding potential altogether needs to be conducted” because it would not further strengthen the conclusion of this study. In a previous study, the Goldberg group nicely showed that induction of a *KLF4ΔC* (lacking the entire KLF4 DNA binding domain) in mouse retinal ganglion cells could no longer suppress the axon outgrowth as the WT did (Moore et al., *Science* 2009 326(5950):298-301). Similarly, induction of the *R458A* mutant on the same background showed the same phenotype as *KLF4ΔC* (Goldberg; personal communication), suggesting the removal of the entire DNA binding domain of KLF4 did not cause any additional phenotype. Even if a truncated KLF4 mutant might cause additional phenotypes, it is not the focus of this study. In our opinion, the R458A mutant is the most relevant comparison with KLF4 WT because it loses the ability to bind to methylated DNA motifs while maintaining binding activity to KLF4’s canonical motif (Hu et al., *eLife* 2013). Using a truncated KLF4 mutant that lacks the entire DNA binding domains would not help us to identify mCpG-dependent KLF4 targets because we would not be able to distinguish KLF4-mCpG binding from the canonical KLF4 binding activity.

*2) Causality of DNA methylation dependent binding of Klf4 on the genome and induction of some target genes needs to be more fully established. As pointed by reviewer #3, the presence of the binding within 10kb of the promoter does not prove direct targeting relationship. New experimental data should be provided to support the notion that a target gene is directly regulated by Klf4's binding to the methylated DNA near the gene. This could be CRISPR/cas9 mediated genome editing or other means.*

We thank the reviewers for this constructive suggestion. On the basis of the reviewers’ suggestion, we selected two genes, namely *RAC1* and *RHOC*, because 1) they are known to play a crucial role in cell migration; 2) we found that they were associated with WT-specific ChIP-seq peaks and genome-wide bisulfite sequencing confirmed that these peaks contained highly methylated CpGs; and 3) these peaks are in close proximity to the coding regions of *RAC1* (-922 bp to TSS) and *RHOC* (-507bp to TSS). In a newly added case of *RAC1*, we first showed that induction of KLF4 WT, but not the R458A mutant, induced *RAC1* expression at both mRNA and protein levels using RT-PCR and immunoblot analysis, respectively (Figure 5—figure supplement 1). Using a primer pair flanking *RAC1* ChIP-seq peak, we confirmed that this fragment could be ChIPed substantially better with the KLF4 WT than the R458A mutant (Figure 5—figure supplement 1 C). To further establish the causality of DNA methylation-dependent binding of KLF4 on induction of its target genes, we performed bisulfite sequencing against this *cis*-regulatory element with and without the 5-Aza treatment. We observed that the methylation level of four of the six CpGs was reduced from ~100% to an undetectable level after 5-Aza treatment (Figure 5—figure supplement 1). Most importantly, removal of methylation almost completely abolished KLF4 WT binding to this *cis*-regulatory element (Figure 5—figure supplement 1), and expression of *RAC1* was also significantly repressed (*p* < 0.001; Figure 5—figure supplement 1).

In the case of *RHOC*, we carried out additional assays to further establish the causal relationship. We performed ChIP-PCR against its *cis*-regulatory element (-507 bp to TSS) with and without 5-Aza treatment and observed that reduction of methylation in this region also substantially reduced recruitment of KLF4 WT. Taken together, these new results demonstrate that KLF4 WT activated the expression of these two genes via binding to their highly methylated *cis*-regulatory elements nearby the coding regions. The new data is added as panel Figure 5 and Figure 5—figure supplement 1).

To examine the impact of DNA demethylation on the global expression level, we performed RNA-seq analysis and compared changes in gene expression profiles with and without 5-Aza treatment. We observed that in the absence of KLF4 WT induction the overall expression levels of the 116 KLF4 WT-specific target genes were not significantly affected by 5-Aza treatment (Figure 4), as expected. However, when KLF4 WT was induced, the expression level of 101 of the 116 genes was reduced in 5-Aza treated cells as compared with untreated cells (Figure R4, right panel; *p* = 1.2e-14 compared to all expressed genes), suggesting that DNA methylation is responsible for the transcription activation of these 101 genes. The new data is in Figure 4.

In our opinion, using CRISPR/Cas9-mediated genome editing may not help strengthen the causality because the mCpG-dependent KLF4 DNA binding activity is also sequence-specific (Hu et al., *eLife* 2013). In other words, the use of CRISPR/Cas9 to mutate the DNA sequence (i.e., CpG sites) of interest would not assist us to de-couple the effects of DNA sequence and DNA methylation on the KLF4 binding activity. Ideally, a CRISPR-guided site-specific DNA demethylation approach would be appropriate; however, this method might not guarantee maintenance of low DNA methylation level due to the endogenous DNA methyltransferase activity.

*3) The treatment of cells by 5-Aza is inducing global DNA demethylation, and the observed phenotypes in the cell system can be attributed to non-specific effects. The authors need to discuss this aspect of the experiments to avoid drawing un-substantiated conclusions.*

We agree with the reviewers and have added the following discussion to address this concern in Discussion section.

“We are fully aware that 5-Aza is only a sub-optimal approach to examine the effect of specific methylation sites, because it induces global demethylation and may have some non-specific effects (Christman et al., 2002). However, we have multiple lines of evidence showing that the observed phenotypes are due to specific interactions between KLF4 and methylated sites. (1) In our control experiments when KLF4 R458A cells were treated with 5-Aza, we did not see changes in gene expression and cell migration. (2) In the focused studies of *RHOC* and *RAC1*, we showed that DNA methylation is responsible for KLF4 binding and gene activation. *RHOC* and *RAC1* are major GTPase involved in cell migration and motility. (3) Comparison of ChIP-seq between KLF4 WT and R458A showed that WT-specific binding sites are enriched for highly methylated sites. Given the phenotypic differences between KLF4 WT and R458A, the result suggested that the observed phenotypes in WT are likely due to specific interactions between KLF4 and DNA methylation sites. (4) Our new RNA-seq data in the presence of 5-Aza also indicated that 5-Aza globally reversed the up-regulation of many KLF4-mCpG targets.”

*4) Statistical methods need to be better described and experimental data need to be more rigorously interpreted. Please check their comments carefully and make sure to remove or modify the particular statements in the revised manuscript.*

We thank reviewers for the constructive suggestion on the statistics analyses. We have added more rigorous statistical analyses throughout the revised manuscript according to the reviewers’ suggestions. Detailed responses can be found in the specific points raised by each reviewer. Please refer to our responses to reviewer #1’s major points 1 and 2; reviewer #2’s major points 1, 2, 3 and 7; reviewer #3’s major points 1 and 2. All of these new analyses have been incorporated into the revised manuscript.

*Reviewer #1*

*[…] Major problems:*

*1) Differential binding of WT and R458A Klf4 in GBM cells: Is this due to failure of the mutant protein to bind methylated DNA? Unfortunately, the evidence presented is weak: the authors performed Bisulfite Sequencing on 15 sites and found that 10 of them contain methylated CpG. Are these DNA sequences recognized by WT Klf4 but not mutant klf4? EMSA should be performed to establish this point. Further, 15 is too small a number to generalize to the rest of the differential binding sites. It is important to carry out base-resolution analysis of DNA methylation in these cells and determined if differential DNA binding by WT and R458A Klf4 in the GBM cells is indeed correlated with presence of DNA methylation at the Klf4 binding motifs in these regions.*

We appreciate the reviewer’s insightful suggestion. We have performed the whole genome bisulfite sequencing in GBM cells and combined this new dataset with KLF4 WT and R458A mutant ChIP-seq datasets. Please see our responses to essential revision #1 and #2 above for more details. Here are some highlights that are relevant to answer the reviewer’s questions: 1) WT-specific ChIP-seq peaks contained CpGs of a much higher methylation level than those peaks shared by the WT and R458A mutant (Figure 3) DNA motif analyses against those WT-specific peaks identified 10 significantly enriched motifs, all of which contained at least one CpG with significantly higher methylation level than those identified among the shared peaks (Figure 3). More importantly, in-depth in vivo studies demonstrated that removal of DNA methylation at the *cis*-regulatory elements in the promoters of two cell migration genes (*RAC1* and *RHOC*) abolished the recruitment of KLF4 WT to these promoters and repressed their gene transcription (Figure 5—figure supplement 1). Taken together, these new results demonstrated that KLF4 WT activated the expression of these two genes via binding to their highly methylated *cis*-regulatory elements nearby the coding regions.

*2) The authors claimed that 116 genes are activated by mCpG-dependent KLF4 binding activity. Again, the evidence to support this claim is weak. Besides the reasons outlined in #1, there is also a need to establish the causality of KLF4 binding to methylated DNA in the cells and activation of these genes. Since the authors observed partial loss of cell adhesion and migration after 5-AZA treatment, it is important to show that the induction of these 116 genes is affected by 5-AZA in the direction that is consistent with their being targeted by Klf4.*

The reviewer suggested a very important way to strength our findings. Please see our responses to essential revision #2 above for more details. To examine the impact of demethylation on global expression level, we performed RNA-seq analysis and compared changes in gene expression profiles with and without 5-Aza treatment. We observed that in the absence of KLF4 WT induction the overall expression levels of the 116 KLF4 WT-specific target genes were not significantly affected by 5-Aza treatment (Figure 4), as expected. However, when KLF4 WT was induced, the expression levels of 101 of the 116 genes were significantly reduced in 5-Aza treated cells as compared with untreated cells (Figure 4 B, *p* = 1.2e-14 compared to all genes), suggesting that DNA methylation is responsible for the transcription activation of these 101 genes.

*3) The 5-AZA treatment leads to partial loss of migration of cells expressing WT Klf4. Is this due to DNA methylation dependent binding of Klf4? It is necessary to demonstrate loss of DNA methylation and loss of Klf4 binding at some sites after treatment.*

The answer is yes. Please see our response to essential revision #2.

On the basis of the reviewers’ suggestion, we selected two genes, namely *RAC1* and *RHOC*, because 1) they are known to play a crucial role in cell migration; 2) we found that they were associated with WT-specific ChIP-seq peaks and genome-wide bisulfite sequencing confirmed that these peaks contained highly methylated CpGs; and 3) these peaks are in close proximity to the coding regions of *RAC1* (-922 bp to TSS) and *RHOC* (-507 bp to TSS), respectively. In the case of *RAC1*, we first showed that induction of WT, but not the R458A mutant, could induce RAC1 at both mRNA and protein levels using RT-PCR and immunoblot analysis, respectively (Figure 5—figure supplement 1). Using a primer pair flanking *RAC1* ChIP-seq peak, we confirmed that this fragment could be ChIPed substantially better with the KLF4 WT than the R458A mutant (Figure 5—figure supplement 1). To further establish causality of DNA methylation-dependent binding of KLF4 on induction of its target genes, we performed bisulfite sequencing against this *cis*-regulatory element with and without the 5-Aza treatment. We observed that the methylation level of four of the six CpGs was reduced from ~100% to an undetectable level after 5-Aza treatment (Figure 5—figure supplement 1). Most importantly, removal of methylation almost completely abolished KLF4 WT binding to this *cis*-regulatory element (Figure 5—figure supplement 1), and expression of *RAC1* was also significantly repressed (*p* < 0.001; Figure 5—figure supplement 1). The same set of assays was also carried out to analyze the *cis*-regulatory elements of *RHOC* and the same results were obtained. Taken together, these new results demonstrated that KLF4 WT activated the expression of these two genes via binding to their highly methylated *cis*-regulatory elements nearby the coding regions.

*Reviewer #2:*

[…] Major points:

*1) The authors observe KLF4 ChIP-seq in WT and R458A transfected cells. There are some key aspects that need to be included. How many / percentage of canonical and methylated sites are specific or shared between WT and R458A cells. A representation of the motifs needs to be addressed. Figure 3, the authors need to provide the input track to the screenshot; it seems to be the background is high. Some other similar screenshots of genes as supplementary figures will be useful. MACS 1.4 has been used to call peaks for KLF4 while again MACS2 has been used to call peaks while comparing with the acetyl mark. It would be better if the authors use the same pipeline to call peaks, as they are different. Why do the authors need to call a broad peak in this case?*

We appreciate the reviewer’s positive comments and insightful suggestions to strengthen our findings. As discussed in essential point #1 above, we have performed the whole genome bisulfite sequencing and combined this dataset with our ChIP-seq datasets. We observed that 10 methylated 6-mer motifs, each containing one CpG of high methylation level (β>0.6), were significantly enriched in the KLF4 WT-specific peaks, while distinct motifs without CpG sites were found enriched in the shared ChIP-seq peaks. Please see essential point #1 for more details.

We calculated numbers of WT-specific and shared peaks, respectively, including one of non-methylated motifs (5’-AAAGGA) and methylated (5’-CCCGCC) sites. We found that the non-methylated motif was significantly overrepresented in shared peaks (fold enrichment = 2.9, p = 7.7e-41), whereas the methylated motif was significantly enriched in WT-specific peaks (fold enrichment = 2.4, p = 3.0e-139).

We have also provided the input track and screenshot of some KLF4-mCpG target genes (see Figure 3—figure supplement 1). The new data is now in Figure 3 and Figure 3—figure supplement 1.

We agree that MACS2 is widely used for calling ChIP-seq peaks. In general, the results by MACS 1.4 were quite similar to those obtained using MACS2. However, in our case some peaks observed obvious to eye-balling (for example, the left one of twin peaks associated with RHOC in Figure 3) could not be identified by MACS2. Therefore, we decided to use MACS1.4 for better sensitivity. For histone modifications in this paper, we used MACS2 to call broad peaks without any problem.

*2) The R458A mutant binds to ~1200 regions, the authors need to provide details regarding these regions.*

We now provide the chromosomal locations of the WT and R458A shared peaks listed in Figure 3—source data 33.

*3) The methods do not indicate whether biological replicates were performed for the sequencing based assays, in that case reproducibility is questionable. The authors can also provide a supplementary summary table for the mapped reads for all the sequencing experiments.*

We appreciate the reviewer’s comments. We have added the summary tables in [Supplementary-material SD1-data] and 2 for the mapped reads for all the sequencing experiments in our supplementary table and analyzed correlation of RNA-seq reads between the biological replicates. As showed in Figure 3—figure supplement 1below, the reproducibility is very high (c.c. > 0.8) between our two experiments. The new data is now in Figure 3—figure supplement 1.

*4) Figure 1: KLF4 expressed at a much higher level in the mutant version and can be observed consistently across the doses, still the authors find loss of physiological functions and KLF4 binding peaks needs explanation. The authors need to include a negative control or some KLF4 silencing assay to show WT KLF4 is directly correlated to the observed physiological changes such as cellular migration / wound healing / adhesion.*

We apologize for the poor explanation of the cell system we used in our study. The purpose of using tet-on system, i.e., doxycycline (Dox)-inducible cell lines for KLF4 functional study, was to have an internally-controlled system, such that when the cells are not treated with Dox, they behave like the parental cells, therefore serves as a control cell line to study the biological function of KLF4. Compared to their own control cells, the induction level of KLF4 R458A was a bit higher than that of KLF4 WT, consistent with as our more quantitative RNA-seq data indicating the fold change of KLF4 WT and KLF4 R458A are 20 and 27 fold, respectively. However, the differences in expression level did not affect the ability of KLF4’s binding to methylated CpGs, hence cellular phenotype changes. In the revised version, we specifically explained the tet-on cell system we used (Results section). Moreover, as the reviewer suggested, we did cell migration assays in the non-transfected U87 cells (control) and found that non-transfected U87 cells behave the same as the non-Dox treated U87 KLF4 WT cells. Please refer to our response to reviewer #2, major point 8 later.

*5) Figure 2: Comparatively lesser focal location of F-actin and Vaculin can also be noted in R458A cells it will be helpful to if the authors show fields with similar number of cells as that of the WT cells.*

In the revised version, we have chosen more representative photographs to replace Figure 2. These images support our conclusion that after Dox treatment KLF4 WT induced more focal locations as shown by F-actin and vinculin staining, and the cells were more spread out.

*6) Figure 5 subsection “KLF4-mCpG interactions activate RHOC via chromatin remodelling” It is evident treatment of WT cells with 5-aza and doxycycline induces expression of RHOC, so it is apparent expression of RHOC is not absolutely methylation / KLF4 dependent. The authors should explain this in their text. Authors should have R458A version of the figure.*

We redid the WB experiments together with KLF4 R458A mutant cells as the reviewer suggested. It showed that 5-Aza itself had minimal effect on the baseline expression of *RHOC* expression (Figure 5). In KLF4 WT cells, 5-Aza blocked *RHOC* upregulation induced by KLF4 WT. In KLF4 R458A cells, 5-Aza had no effect on *RHOC* expression under similar conditions. The data is now in Figure 5 in the revised manuscript.

*7) What is the methylation status of the WT specific KLF4 chip peaks sharing acetylation mark.*

The reviewer raised a very good point. Of 1,773 WT-specific KLF4 chip peaks sharing acetylation mark, 978 (55.2%) contain CpG sites of >60% methylation level.

*Reviewer #3:*

*[…] Major points:*

*1) […] In addition, two technical modifications of these analyses are needed to further avoid bias. First, the comparisons among binding sites, induced genes, marks etc need to be restricted to genes that are expressed at all in these cells – it is possible to get high overlaps among unrelated regulators only because everything tends to pile up on open chromatin and active promoters. Similarly, the GO analyses later in the paper should be done taking the induced genes as a background. KLF4 overexpression induces migration and morphological changes, which could explain why genes related to this process are expressed. It is possible that KLF4 only directly induces a small number of genes, and the rest of what we are seeing is a secondary effect.*

We appreciate the points raised by the reviewer and performed additional statistical analyses according to the reviewer’s suggestions. We have calculated the statistical significance of the overlapped genes. We focused on 12,824 genes that were expressed (FPKM > 0.5) in U87 cells. Among them, 308 (2.4%) up-regulated genes were identified. In the meantime, we observed that KLF4 mCpG-dependent binding peaks were associated with 1,072 expressed genes, of which 65 were up-regulated genes. The ratio of 6.1% is significantly higher than the ratio of expressed genes not associated with KLF4 binding sites (fold enrichment = 2.5; *p* = 1.2e-12). Adding KLF4 mCpG binding sites in enhancer region, we finally identified 116 targets out of 2,601 genes, indicating significantly over-representation (fold enrichment = 1.9; *p* = 5.0e-13). In the revised manuscript, we added this new analysis in the text (subsection “Identification of direct targets of mCpG-dependent KLF4 interactions in GBM cells”). In addition, we performed whole genome bisulfite sequencing, integrated it with the ChIP-seq datasets, and re-analyzed the data (see our response to essential point #1). We also obtained RNA-seq data after 5-Aza treatment and re-analyzed the entire dataset (see our response to essential point #2). To firmly establish the causality between mCpG-dependent KLF4 binding and target gene activation, we performed in-depth assays to examine impact of de-methylation on the recruitment of KLF4 WT to the promoter regions of two target genes in conjunction with changes in their RNA level (see our response to essential point #2).

*2) It seems possible that global DNA demethylation could attenuate the ability of cells to do many things, and thus the results shown in Figure 2 could be nonspecific. Are there other perturbations that can trigger the same responses that are being assayed, and are these responses intact in aza-treated cells? If not, then the effects cannot really be attributed to a specific effect of KLF4 binding.*

Please see our response to essential point #3. As far as we know, 5-Aza treatment is the only effective means to reduce global DNA methylation level. In response to essential point #3, we have added discussion of 5-Aza treatment to the manuscript. To examine the impact of DNA demethylation on the global expression level, we performed RNA-seq analysis and compared changes in gene expression profiles with and without 5-Aza treatment. We observed that in the absence of KLF4 WT induction the overall expression levels of the 116 KLF4 WT-specific target genes were not significantly affected by 5-Aza treatment (see our response to essential point #2).

*3) I do not see that Figure 5 adds anything to the paper. It is simply a reiteration of assays already used earlier on a global scale, but using PCR assays. The same issues I describe above in major points 1 and 2 also apply to these results – I believe that the final statement "Therefore, high methylation level in RHOC promoter is essential for KLF4 WT to activate RHOC transcription" is rather an overstatement. The first sentence is better supported: "Several lines of evidence supported that DNA methylation mediated RHOC activation by KLF4 WT". On the whole, however, this is the same argument that has already been made in genome-wide analyses, with no new evidence. I would suggest that this section is dispensible, since it only confirms what would be expected from the rest of the manuscript.*

We respectfully disagree with the reviewer. Figure 5 is not redundant because it serves four purposes. First, it addressed the causality issue raised in essential point #2. Second, focused studies are essential for rigorous validation of high throughput analyses, which often contains many false positives. Third, RHOC plays a crucial role in cell migration/adhesion pathways. The discovery that the gene was activated via mCpG-mediated KLF4 binding activity is significant and of great interest in cancer biology. Most importantly, results showed in Figure 5 established a connection between dynamic changes of histone modifications and mCpG-dependent KLF4 binding activity in cells, which offered new molecular mechanism underlying mCpG-dependent gene activation. To address the concern of overstatement raised by the reviewer, we now offer more experimental and computational evidence as described in our responses to essential points #1 and #2. We believe that our in-depth study as illustrated in Figure 5 is well justified.

*4) The association of active marks with induced promoters and with many of the Klf4 sites is unsurprising – this would be seen even if the induction of the supposed "target genes" was indirect, and the KLF4 binding was coincidental. Indeed, the fact that there are 3,593 novel H3K27ac peaks, but only 274 overlap KLF4, shows that the two are largely dissociated. Thus, the statement in the Discussion that "Our study reveals that mCpG-dependent binding activity of KLF4 serves as the link between methylated CpG and changes in chromatin status" is not supported by the data shown. It is "a" link, but not "the" link.*

We have changed this sentence to "Our study reveals that mCpG-dependent binding activity of KLF4 *serves* as a link between methylated CpG and changes in chromatin status" (Discussion section, paragraph five).